# Presence of male mitochondria in somatic tissues and their functional importance at the whole animal level in the marine bivalve *Arctica islandica*

Cyril Dégletagne [1,2✉], Doris Abele[1], Gernot Glöckner [3], Benjamin Alric[2], Heike Gruber[4] & Christoph Held[1]

Metazoans normally possess a single lineage of mitochondria inherited from the mother (♀-type mitochondria) while paternal mitochondria are absent or eliminated in fertilized eggs. In doubly uniparental inheritance (DUI), which is specific to the bivalve clade including the ocean quahog, *Arctica islandica*, ♂-type mitochondria are retained in male gonads and, in a few species, small proportions of ♂-type mitochondria co-exist with ♀-type in somatic tissues. To the best of our knowledge, we report, for the first time in metazoan, the natural occurrence of male and female individuals with exclusively ♂-type mitochondria in somatic tissues of the bivalve *A. islandica*. Mitochondrial genomes differ by ~5.5% at DNA sequence level. Exclusive presence of ♂-type mitochondria affects mitochondrial complexes partially encoded by mitochondrial genes and leads to a sharp drop in respiratory capacity. Through a combination of whole mitochondrial genome sequencing and molecular assays (gene presence and expression), we demonstrate that 1) 11% of individuals of an Icelandic population appear homoplasmic for ♂-type mitochondria in somatic tissues, 2) ♂-type mitochondrial genes are transcribed and 3) individuals with ♂-type mitochondria in somatic cells lose 30% of their wild-type respiratory capacity. This mitochondrial pattern in *A. islandica* is a special case of DUI, highlighted in individuals from both sexes with functional consequences at cellular and conceivably whole animal level.

[1] Alfred-Wegener-Institut Helmholtz-Zentrum für Polar- und Meeresforschung, Bremerhaven, Germany. [2] Univ Lyon, Université Claude Bernard Lyon 1, CNRS, ENTPE, UMR5023 LEHNA, Villeurbanne, France. [3] Institute for Biochemistry I, Medical Faculty, University of Cologne, Cologne, Germany. [4] Max Planck Institute for Evolutionary Biology, Department of Evolutionary Theory, Plön, Germany. ✉email: cyril.degletagne@gmail.com

The bulk of genetic information is stored in the nuclear genome, but all eukaryotes possess mitochondria, small organelles that are equipped with an additional, extra-nuclear genome, unless secondarily lost[1,2]. The genes encoded by the mitochondrial DNA (mtDNA) sequence are finely tuned to the cellular and nuclear background of their "host" cell, which supports and conserves mitonuclear interactions and mitochondrial biology. Different branches in the eukaryote tree are characterized by a variety of mitochondrial inheritance patterns[3], but metazoans adhere almost uniformly to a common pattern characterized by a single, maternally inherited lineage of ♀-type mitochondria[4,5].

Unfavorable mutations can cause cellular energetic imbalance, accelerate aging, and are associated with pathological phenotypes[5–8]. Several processes contribute to mtDNA sequence conservation: (1) negative selection against mtDNA variants (mitotypes)[5]; (2) reduction of the number of mtDNA copies per cell during early embryogenesis[9]; and (3) the non-Mendelian inheritance of mitochondria exclusively through the oocyte (♀-type) but not the sperm. Collectively, these mechanisms reduce the risk of mitonuclear genomic incompatibilities and the propagation of cytoplasmic selfish elements[9–11].

Occasionally, paternal mitochondria are transmitted to off-spring, leading to the simultaneous presence of ♀-type and paternal mitochondria in the same cell. The effects of this mitochondrial heteroplasmy range from relatively mild changes[12] to severe physiological disorders in humans or mice[13–15].

In more than 100 bivalve species, mitochondrial heteroplasmy is the norm rather than the exception[16]. Doubly uniparental inheritance (DUI) is characterized by the co-existence of two distinct sex-associated mitochondrial genomes in a growing number of dioecious bivalve species belonging to the orders Venerida, Unionida, Nuculanida, and Mytilida[16–19]. In DUI species, the reported differentiation between the two mitochondrial lineages ranges from 20% in many marine taxa (orders Mytiloida and Veneroida) and can reach >50% in freshwater mussels (order Unionoida) in their nucleotide sequences[16,18,20,21]. This peculiar mechanism, in which both ♀- and ♂-type mitochondria are transmitted through oocyte and sperm, respectively, was restricted to dioecious bivalves[22]. Recently, Lubośny and collaborators reported a case of DUI in the marine hermaphroditic mussel Semimytilus algosus[23].

Ghiselli and collaborators[11] used immunohistostaining of maternal (F-type) and paternal (M-type) mitochondrial proteins across different stages of germ and somatic cell development in the Japanese carpet shell clam, Ruditapes philippinarum, to unravel the molecular momentum of sex differentiation in a DUI species. It resulted that eventual sex differentiation happens through a meiotic drive that eliminates M-type mitochondria in female germ cells ("late oocytes") and F-type mitochondria in sperm cells. Staining showed primordial stem and germ cells to be heteroplasmic in both sexes, whereas post-meiotic germ cells are strictly homoplasmic for F-type (♀-type) mitochondria in females and M-type (♂-type) mitochondria in male clams. This leads to the development of females which are usually homoplasmic with only ♀-type mitochondria in somatic and gonadic cells, whereas male R. philippinarum, commonly heteroplasmic in soma, sometimes with the ♂-type dominating the ♀-type mitochondria in different tissues, and exclusive ♂-type mitochondria in sperm[15,17,24–27]. Only rarely females, heteroplasmic in somatic tissues, were encountered[27]. The presence of M-type in male somatic tissues is further known to occur in Venustaconcha ellipsiformis and Utterbackia peninsularis[15], and also in Mytilus galloprovincialis[28] and might be more common across DUI species than previously conceived.

The physiological importance of ♂-type mitochondria has so far been demonstrated only for sperm[29], but so far the functional importance of the ♂-type mitochondria in somatic tissues remains unknown.

Interestingly, ♀-type mitochondria can also partly or completely replace ♂-type mitochondria in male gonads as shown for some members of the Mytilidae family, a process called "masculinization"[18,30]. Masculinization of ♀-type mitochondria was shown to improve mitochondrial energetic functions in male gonads[29,31], intuitively suggestive of its preferred inheritance over ♂-type mitochondria.

Here, we report different somatic tissues containing exclusively ♂-type but no ♀-type mitochondria in several males and, even more surprisingly, female individuals belonging to the Icelandic population of the DUI species Arctica islandica[32]. This is also the demonstration of the measurable functional significance of the persistence of somatic ♂-type mitochondria at the whole organism level.

In order to avoid confusion caused by the alliteration of M-mtDNA (i.e. male) and maternal DNA, we refer to the two types of mitochondrial DNA occurring in Arctica islandica as the ♀- and ♂-type mitochondria. The ♀-type mitochondria (called "normal" in the previous article[32]) are normally found in somatic tissues of both sexes as well as in the gonads of females whereas the rarer ♂-type mitochondria ("divergent" in the previous article[32]) are originally found only in male gonads and sperm. The terms male and female are used to denote the sex of the bearer.

## Results and discussion

**The two mitochondrial genomes in A. islandica.** In this study, we sequenced the full ♂-type mitochondrial genome of 16 individuals and obtained a circular molecule of around 17.4 kb, which we compared to the ♀-type mitochondrial genome of three homoplasmic individuals from Iceland: two newly sequenced in this study and one previously published[33]. The phylogenetic analysis of all concatenated protein sequences encoded in the ♂- and ♀-type mitochondrial genomes, respectively, show that the two mitochondrial genomes are reciprocally monophyletic and each other's closest relative, indicating that divergence emerged within the stem lineage leading to A. islandica itself (Fig. 1). Doubly uniparental inheritance of mitochondrial lineages (DUI) also occurs in the bivalve order Unionida, in which the split of the mitochondrial genomes occurred at the base of the higher taxon. Unlike the situation in Mytilida, Nuculanida, and Venerida, in which the diversified mitotypes form a species-specific cluster (Fig. 1), in the Unionida with DUI all-female mitochondrial genomes are typically closer to each other than either is to any paternally inherited mitochondria and vice versa[16,18]. More recent divergence of the two mitochondrial lineages in A. islandica is congruent with the fact that the two mitotypes exhibit a significantly lower divergence (5.5%) compared to what is usually observed in other bivalve families, in which ♂- and ♀-type mitochondrial genomes appear to have split prior to the radiation of the crown group: Mytilida (21–26%)[34] and Unionida (40–50%)[20].

In A. islandica, ♀-type and ♂-type mitochondria exhibit a similar gene organization, with all genes being present on the "+" strand (Fig. 2A, 2B), the common situation in Veneroidea DUI species[35,36]. The divergent sites distinguishing the two mitochondria are not evenly distributed across the genome but are concentrated in protein-coding genes (atp6, cytochrome b, cox3, nad4, nad4L & nad6, especially), 16 S rRNA and two tRNAs (Ile, Cys), and in two non-coding regions that are difficult to align.

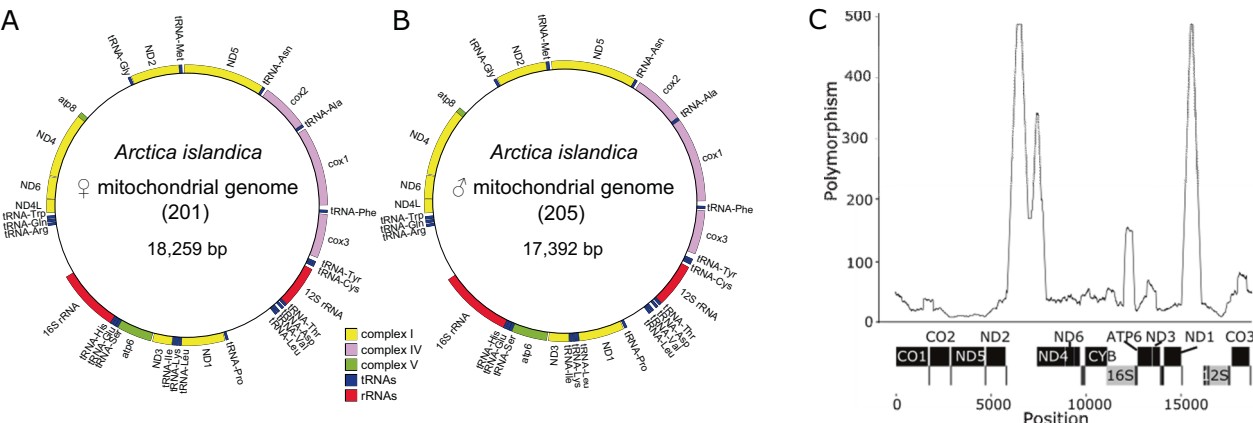

**Fig. 1 Molecular phylogeny of the bivalve subclass Heterodonta based on twelve concatenated mitochondrial protein coding genes (2697 amino acids total).** Numbers indicate the ultrafast bootstrap support (IQTree). Stars represent the most parsimonious position of split events that lead to differentiation into ♂- and ♀-type mtDNAs.

**Fig. 2 *Arctica islandica* mitochondrial genomes.** ♀ (**A**) and ♂ (**B**) mtDNAs and their divergent sites (**C**). Divergent sites were counted in a sliding window of 500 with a step size of 1. More details are available in Supplementary Data 1.

Conversely, most of the tRNAs, the *12 S* rRNA, *cox2*, *nad3*, and *nad5* are more conserved (Fig. 2C). Interestingly, the ♂-type mitochondrial genome is characterized by one cox1 elongation of 10 amino acids and a 127 bp indel in *16 S* rRNA, the presence or absence of which was used to distinguish between ♀- and ♂-type mitochondria in our PCR mitotype assay (Fig. 2., Supplementary Data 1). Although DUI species are frequently characterized by a

3' extension or duplication of their cox2 gene, a feature that some authors suggested to be involved in the transmission of ♂-type mitochondria[16,36], no such modification of the *cox2* gene was found in *A. islandica*.

**The occurrence of ♂-type mitochondria in somatic tissues is limited to subarctic populations.** We observed complete

**Table 1 Location and sampled size of each population used for the phylogeographic analysis.**

| Population | Location | Number of individuals | Number of specimens carrying the ♀ mtDNA | Number of specimens carrying the ♂ mtDNA |
|---|---|---|---|---|
| Norway | 69°39 N, 18°57 E | 33 | 30 | 3 |
| Iceland | 66°01 N, 14°51 W | 60 | 50 | 10 |
| Baltic Sea | 54°32 N, 10°42 E | 77 | 77 | 0 |
| Helgoland | 54°09 N, 07°47 E | 47 | 47 | 0 |
| US | 41°00 N, 71°00 W | 58 | 56 | 2 |
| Kattegat | 56°10 N, 11°48 E | 17 | 17 | 0 |
| White Sea | 66°18 N, 33°38 E | 23 | 23 | 0 |
| | total | 315 | 300 | 15 |

replacement of the ♀- by ♂-type mitochondria in somatic cells of both sexes only in the Northern oceanic populations of *A. islandica*'s distribution range. Using partial *cytochrome b* and *16 S* mtDNA sequences as proxies in 315 individuals from the entire distribution area of *A. islandica* (Table 1), we found that 300 male and female individuals from all sampled populations had the typical metazoan configuration with only ♀-type mitochondria in adductor muscle. In 15 individuals, however, we detected ♂-type mitochondrial DNA in adductor muscle. These individuals (♂-type mitochondria in adductor muscle; Supplementary Figure 1, Table 1) amounted to 11% of the Icelandic, Norwegian, and Northwest Atlantic samples. Despite extensive sampling in the other parts of the distribution area (*n* = 164; Table 1), no further individuals with ♂-type in adductor muscle were detected in populations from shallower shelf seas (Baltic, Kattegat, White Sea, the North Sea around Helgoland), indicating absence or rarity of this condition in these populations. The same ♂-type mitochondrial DNA has already been reported in the somatic tissue of two Atlantic *A. islandica* individuals from Iceland and Nova Scotia[37], who named it haplotype "X" and excluded it from their analysis as a case of inexplicable heteroplasmy. Phylogenetic reconstruction, combining all *cytochrome b* and *16 S* haplotypes from our previous study[32] and this paper, identified two groups, the first grouping the ♂-type mtDNA with the divergent mtDNA haplotypes of male gonads and the "X" haplotype[37], and the second grouping the ♀-type mtDNAs with the common mtDNA haplotypes characteristic of somatic tissues and female gonads (Supplementary Figure 2–3).

Considering the possibility that the ♂-type mtDNA might evolve faster with less strict purifying selection than the ♀-type mtDNA in DUI species[16], it is possible that some cases of somatic ♂-type mitochondria escaped detection in our study if a mutation occurred at the binding sites of our ♂-type mtDNA specific primers. However, the successful amplification of only the ♂-type mtDNA in two different loci (*cytochrome b* and *16 S*) in sperm[32] and in somatic tissue (this paper) indicates that we can readily detect the ♂-type mtDNA if present. The population-specific occurrence of ♂-type mitochondria in adductor muscle, and probably in other somatic tissues, could be the result of a strong selection against ♂-type mitochondria in individuals thriving in areas in which environmental conditions are more variable or alternatively reflect a founder effect even in the absence of selection.

**Exclusive presence of ♂-type mtDNA in somatic tissues of male and female *Arctica islandica*.** To investigate the tissue- and sex-specificity of somatic ♂-type mitochondrial DNA presence, an additional extensive tissue collection (gills, mantle, foot, and adductor muscle) of 181 Icelandic *A. islandica* from the Icelandic shelf was screened using multiplex PCR (see Materials and methods section for details). In total, 18 specimens among 181 individuals were identified which carried the ♂-type

mitochondria in adductor muscle. Ten of these were clearly male whereas a surprising number of six were female, and the remaining two specimens were difficult to sex. Cases of mito-chondrial heteroplasmy with ♂- and ♀-type mitochondria in somatic tissue have previously been reported for several DUI species. In the most extensively studied Mytilidae, some studies report a consistent prevalence of M-type mtDNA (♂) in somatic tissues of males, combined with variable amounts of F-type mtDNA. Also in females over 50% of adductor muscle samples analyzed in the study of Obata and collaborators[25] were hetero-plasmic, with the random occurrence of M-type mtDNA in additional somatic tissues (e.g., foot, mantle). Moreover, in female somatic tissues, the ♀-type was always predominant[15,17,24–27].

To test whether the two mitochondrial types co-occur within each individual and validate our multiplex PCR assay, we performed a quantitative PCR with primer pairs specific for either mitochondrial type from a randomly picked subsample of 21 individuals out of the 163 individuals bearing ♀-type mitochondria and compared them to the 16 sexed individuals carrying ♂-type mitochondrial DNA. Most of these individuals are homoplasmic as the qPCR results revealed only a single somatic mtDNA (either ♂- or ♀-) type. Only six out of 37 individuals turned out to be heteroplasmic for ♀- and ♂-type mitochondrial DNA in at least one of the tissues tested. In all of these potentially heteroplasmic individuals, ♂-type mitochondria were found only in a subset of tissues tested. Their abundances ranged between one and up to two orders of magnitude lower than the ♀-type mitochondria (four individuals in one of the three tissues, ♂-type mitochondria 100 times less abundant; two individuals with heteroplasmy in gills and mantle only, ♂-type mitochondria up to 10 times less abundant). As an independent additional test, we mapped the reads of two Illumina sequencing runs (F-201: ♀-type and M-205: ♂-type, DNA extracted from adductor muscle) against specific ♂- and ♀-type parts of the mitochondrial genome. No reads originating from the ♂-type mitochondria mapped to the ♀-specific region and vice versa (Fig. 3). These results, in combination with the absence of detection of ♀-type mitochondria with predominant ♂-type mitochondria in the foot, gills, and mantle (Fig. 4), suggest the homoplasmic occurrence of the M-type genome in *A. islandica* somatic tissues.

**Transcription of ♂-type mtDNA in somatic tissue alters mitochondrial complex functionality.** The number of mito-chondria per cell is linked to the cellular metabolic requirements of the organism and can be roughly approximated based on the mtDNA copy number[27]. Using the cytochrome b alleles specific for ♀- and ♂-type mtDNA, we showed that mitochondrial density is 2 to 5-fold higher in individuals that possess mostly/only ♂-type compared to those carrying only the ♀-type mitochondria in somatic tissues (Fig. 4). Likewise, male *V. philippinarum*, which carry a mix of ♂- and ♀-type mitochondria in their somatic

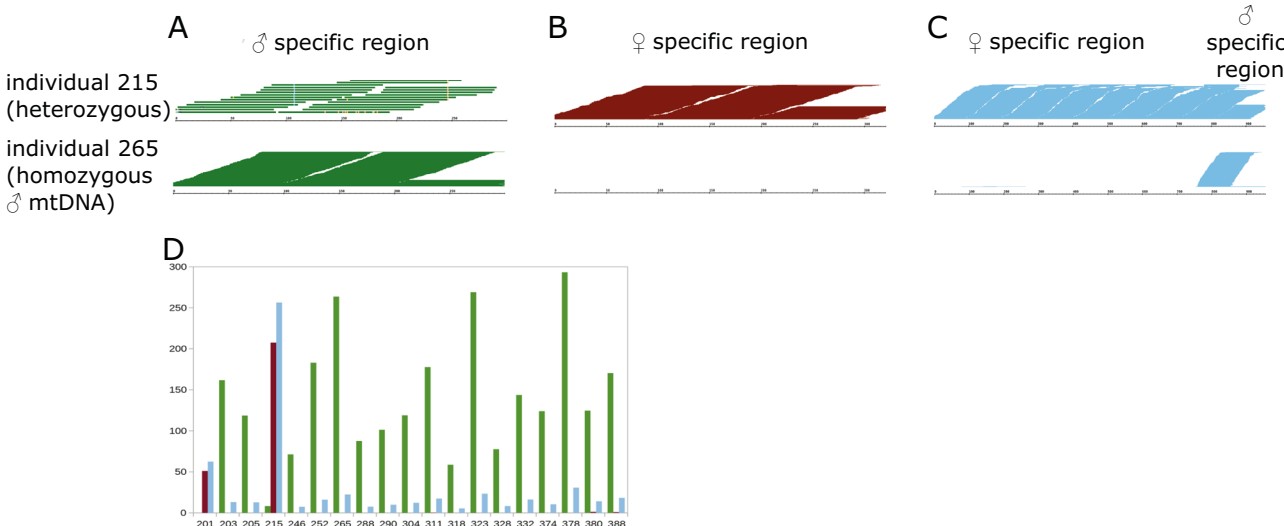

**Fig. 3 Read coverage of specific mitochondrial regions.** Read coverage is depicted for individual 215 (containing ♂ and ♀-type genomes) and individual 265 (containing ♂-type genome only). Region coverage is depicted in green for the ♂-type genome from position 2221 to 2520 (**A**), in dark red for the corresponding region in the ♀-type genome from position 2554 to 2874 (**B**), and in light blue for a ♀-type genome region with only a few ♂-type genome similarities from position 9270 to 10106 (**C**). **D** Comparison of the strains analyzed by number of matching reads per 100 bases. X-axis: strain numbers, Y-axis: counts. Only in strains 201, 215, 311, 380, and 388 reads with ♀-type genome signature were found. Other strains presumably contain ♂-type genomes only in their somatic tissues.

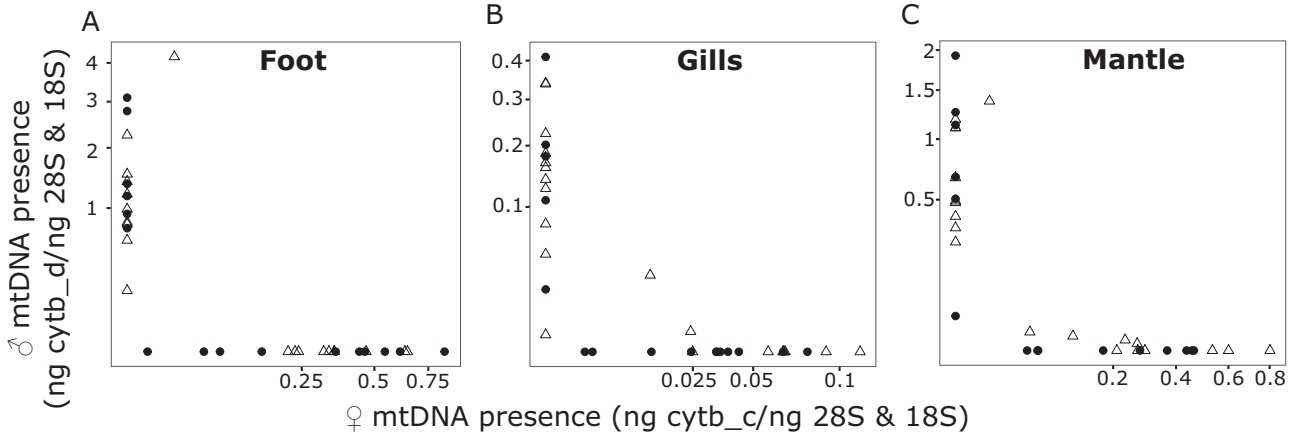

**Fig. 4 Presence of ♂- and ♀-type mtDNAs in *A. islandica* somatic tissues.** The ratio of ♀ (*x*-axis) and ♂ (*y*-axis) *cytochrome b* forms (in ng [nanograms] of *cytochrome b* DNAs divided by the geometric mean of *28 S* and *18 S* DNAs in ng) was determined by a qPCR in the foot (**A**), gills (**B**) and mantle (**C**) for 37 individuals, male (open triangle) and female (filled circle).

tissues, possess more overall mitochondria (♂- and ♀-type) than ♀-type homoplasmic females[27]. These differences in mtDNA copy number either signify compensatory potential for reduced biochemical efficiency of proteins encoded in ♂-type mitochondria, or an increase in the replication rate of the ♂-type mitochondrial DNA due to specific sequences in the mitochondrial control region that might enhance their replication[27].

Using quantitative PCR, we showed that individuals with ♂-type mitochondria exclusively transcribe the corresponding mRNA and vice versa for ♀-type mitochondria bearing individuals (Fig. 5A). Exclusive expression of the ♂-type mitochondrial genes in *A. islandica* (Fig. 5A) causes a reduction in the capacity of complexes co-encoded in the nuclear and mitochondrial genomes compared to specimens carrying only ♀-type mitochondria. ETS complexes I + III in vitro capacity per mg of tissue was reduced by 31% in individuals with ♂-type mitochondria (compared to individuals carrying the ♀-type mitochondria) ($F_{1,31} = 40.58$, $p < 0.001$) regardless of sex (Fig. 5B);

approximately the same reduction was observed for the downstream complex IV (COX) (33% reduction for individuals carrying ♂-type mitochondria, $F_{1,31} = 20.01$, $p < 0.001$) (Fig. 5C). In contrast, the activity of the mitochondrial enzyme citrate synthase from the TCA cycle, which is encoded exclusively in the nuclear genome, depended exclusively on the sex of the individual (42% reduction in female individuals, $F_{1,31} = 10.44$, $p = 0.003$) regardless of the mitotype (Fig. 5D). Non-synonymous substitutions inducing differences in protein amino acid composition between the two mitochondrial types (Supplementary Data 1) might be the cause of the reduced biochemical efficiency of co-encoded mitochondrial enzymes (Fig. 5B–5C).

Recent papers analyzed bioenergetics in spermatozoa, eggs, and gills of DUI species, including *A. islandica*, from the Northeast Atlantic, compared to bivalves relying strictly on maternal inheritance (SMI)[38,39]. This paper was on understanding the evolutionary benefit of M-type mitochondria in DUI sperm for male reproductive success (e.g. sperm motility and conservation

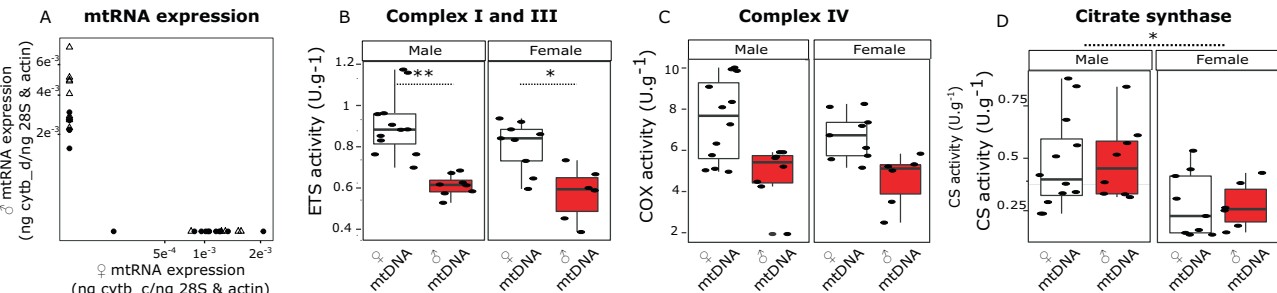

**Fig. 5 Expression of the ♀ and the ♂ mtRNA and their impact on mitochondrial enzyme activities in gills. A** Expression level of ♀ (x-axis) and ♂ (y-axis) cytochrome b forms (in ng of cytochrome b cDNAs per the geometric mean of ng of 28 S and actin cDNAs) measured in gills for 31 individuals, male (open triangle) and female (filled circle). ETS: Electron transport system (**B**) and COX: cytochrome oxidase (**C**) activities in Units per Weight (g) measured in gills *A. islandica* of individuals carrying the ♀ (white bar plot, n = 21) or the ♂ mtDNAs (red bar plot, n = 14). CS: citrate synthase (**D**) activity in Units per Fresh Weight (g) measured in gills *A. islandica* of both sexes previously genotyped (white bar plot: ♀ mtDNA, n = 21; red bar plot: ♂ mtDNA, n = 14). p-value < 0.05 Kruskal Wallis test. *p-value < 0.05, **p-value < 0.01 Wilcoxon Post hoc test.

of sperm that persists in the *offspring to start a "new mitochondrial generation"*). The authors documented "energy saving" functions of DUI sperm mitochondria which only use a smaller fraction of their maximum respiratory capacity (coupled /uncoupled respiration) for sustained motility compared to SMI sperm. Less excess COX activity over the upstream ETS system in DUI sperm mitochondria confers tight control of electron flow and speaks for a less reduced state of ETS intermediates and hence low ROS production compared to F-type mitochondria. In the interpretation of the authors, this supports better conservation of mitochondrial and mt-DNA integrity in sperm of DUI species. In a similar manner, our analysis indicates reduced OXPHOS capacity of ♂-type mitochondria in *A. islandica* somatic tissues (30% reduction of ETS and COX activity) compared to the ♀-type mitochondria of "normal" specimens (note that we did not analyze mitochondrial isolates and cannot – at this stage - comment on mitochondrial respiration and coupling rates).

Combining our results from the Icelandic and other subarctic *A. islandica* populations indicates that individuals can tolerate either form of mtDNA in their somatic cells, presumably only under certain environmental circumstances[11]. The energetically constrained ♂-type mitochondria appear to pose no limitation at constant low water temperatures prevailing around Iceland, where metabolic rates are constrained by the low temperature and presumably also the limited food supply during the Arctic winter. The burrowing lifestyle and the low metabolic rates of these clams support their capacity to rapidly enter a state of metabolic suspension[40,41]. More generally, Moss and collaborators[42] showed that extant high latitudes bivalves grow more slowly and attain longer lifespans than mid-latitude species, and this also applies to the population comparison in *A. islandica*[43,44]. Hence, we conclude that ♂-type mitochondria, while not presenting an explicit advantage, maybe better tolerated at the high latitude by subarctic populations.

**Somatic ♂-type mitochondria may influence female *A. islandica* growth rates.** The fact that animals carrying the ♂-mitotype in somatic tissues constitute around 10% of the individuals within the Subarctic oceanic populations requires additional scrutiny. In our study, between-group PCA indicated 55.5% (simulated p-value = 0.001) of the total variance in sex and mitotype segregation among *A. islandica* populations to be explained by factors "age of the individuals", "biochemical activity" (COX and ETS activities) and "morphological traits" (Length of the Strongest Growth – LSG-, shell - SW - and animal weight - AW; see Fig. 6A). This relationship is congruent with our previous results

showing that the individuals bearing somatic ♂-type mitochondria have lower capacities of mitochondrial complexes per unit tissue weight than the ones carrying somatic ♀-type mitochondria. To explore the impact of mitotypes on morphological parameters, we compared LSG (Fig. 6B), AW (Fig. 6C), and SW (Fig. 6D) over the age of the individuals, for all specimens available in our dataset (145 specimens for which all parameters were measured). Because these morphological parameters are influenced by age, we used nonlinear models to represent our data, and extracted standardized residuals (Supplementary Figure 4). Curiously, we found that females with somatic ♂-type mitochondria tend to have higher standardized residuals for all parameters tested, which might indicate that they are heavier and have a bigger and heavier shell than those with ♀-type somatic mitochondria. However, at present such data have to be used with caution as the evidence is only based on six females with somatic ♂-type mitochondria, and these individuals do not fall outside the range of confidence intervals for LSG, AW or SW observed for the other individuals. This tendency would be highly difficult to prove, even with extra sampling, as females carrying somatic ♂-type mitochondria are rare (only 6 ♂-type mtDNA females found in 166 sexed individuals). Moreover, assessing the impact of the mitotype alone would require all individuals to grow under identical conditions, which requires a laboratory set-up for centenarian bivalves. As fecundity generally increases with size in long-lived bivalves[45], bigger females with heavier shells can be expected to contribute disproportionally to producing the next generation of offspring. The presence of somatic ♂-type mitochondria may hence be favored in species with lifelong reproductive activity such as the sedentary bivalve *A. islandica*. We propose longevity extremes such as 507 years of lifespan[46] to require a very special disposition even in a centenarian species.

## Conclusion

Here we describe a clear example in DUI species with a complete replacement of maternally inherited by paternally inherited mitochondria in somatic tissues. The predominant presence of ♂-type mitochondria in somatic cells might be caused by a failure of the DUI-specific compartmentalization of mitochondria in the embryo. This unprecedented case of individuals carrying only or mostly ♂-type mitochondria in somatic tissues represents a previously undescribed condition of mitochondrial-nuclear crosstalk in DUI species, which may either be tolerable or even confer adaptive advantages in specific environments. It alters mitochondrial ETS complex capacities, and has the potential to influence life-history traits especially in female bivalves. It fits with the strategy of "life at the slow lane" attained by high latitude

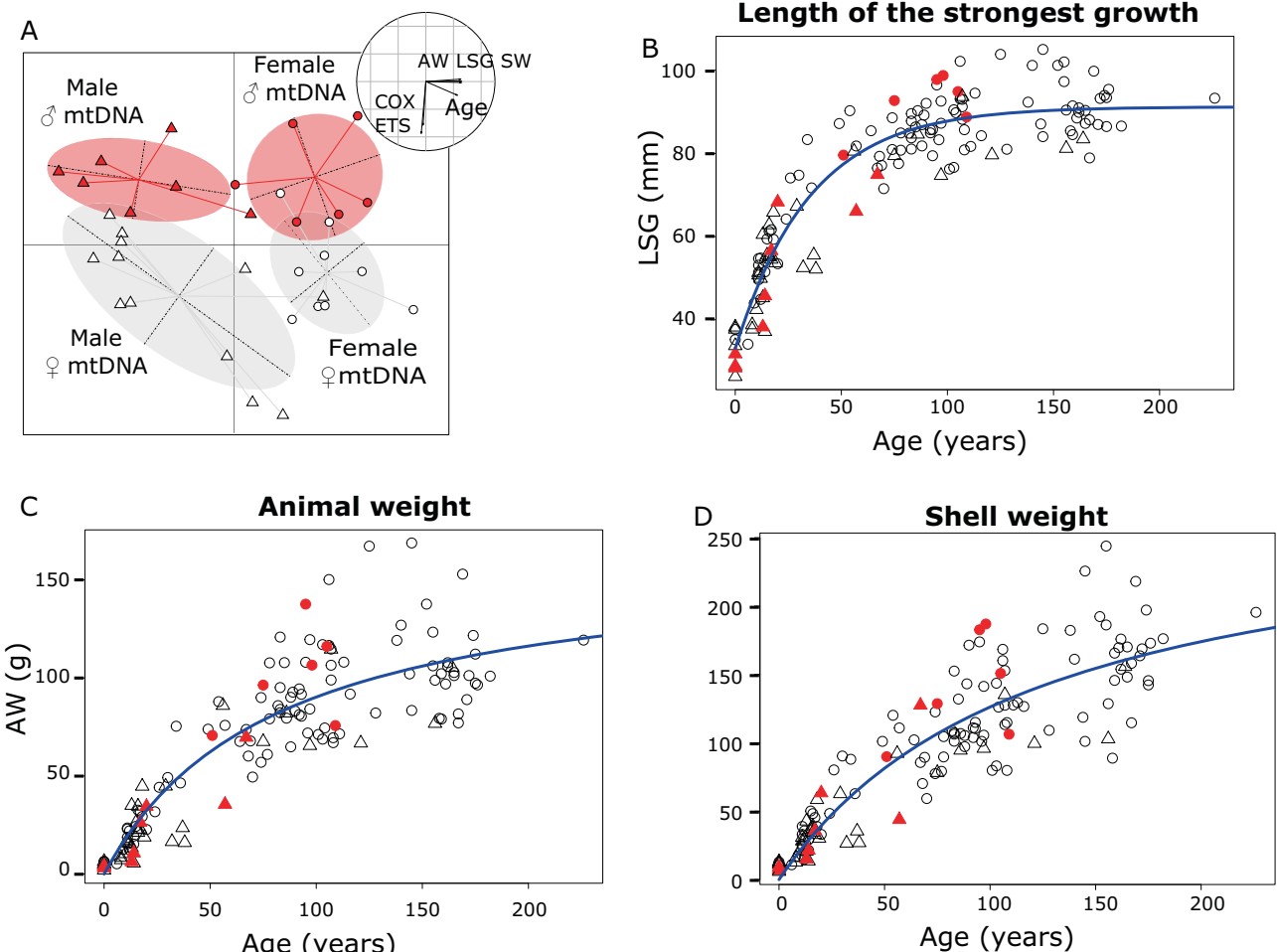

**Fig. 6 Impact of the presence of the mtDNA type on individual morphological parameters.** Between-groups PCA (**A**) is able to distinguish *A. islandica* specimens according to their sex (circle: female, triangle: male) and previously determined mitotype (open symbols ♀ mtDNA, n = 16, red symbols: ♂ mtDNA, n = 15). The circle at the top-right of the plot describes the ecological and physiological meaning of axes 1 and 2 of between-groups PCA: COX, cytochrome oxidase; ETS, electron transport system; AW, animal weight; SW, shell weight; LSG, line of the strongest growth. The effect of the mtDNA haplotype (open symbols: ♀ mtDNA, n = 124; filled red symbols: ♂ mtDNA, n = 16) on morphological parameters, length of strongest growth (**B**), animal weight (**C**) and shell weight (**D**) was represented over the age of the individuals The blue line represents the predicted value of the fitting model for length of strongest ($y = 91.208-58.125e^{-0.028x}$), animal weight ($y = 2.017x/(1 + 0.012x)$) and shell weight ($y = 2.296x/(1 + 0.008x)$).

populations thriving in the permanent cold. This is in stark contrast with the severe physiological consequences of paternal mitochondrial inheritance in mice or human[12–14].

The correlation between mtDNA mutations, mitochondrial functional deficiency, and the occurrence of several debilitating and lethal diseases have recently garnered strong interest in clinical science[5,6,47]. Advances in the replacement of mutated mtDNA in oocytes by a healthy version are raising hopes for new treatments to be developed. At the same time, this gives cause to concerns with respect to the impact such replacements might entail for cellular function, especially regarding the consequences of the potential disruption of the co-evolution between the nuclear and the mitochondrial genomes[48]. Further investigations into natural cases of mtDNA replacement are promising to generate new knowledge of the mechanisms necessary for the successful interaction of nuclear genotypes with alternating mitochondrial counterparts. They will open new doors within the mtDNA conservation described in metazoans, the inter-disciplinary field of mitonuclear ecology[49,50], and the general role of mitonuclear compatibility in organismal adaptation and speciation.

## Methods

**Arctica islandica.** As the study deals with and abundant and nonendangered marine invertebrates that are also commercially used in Norway, no ethical approval was needed for this work. Furthermore, we would like to state here that all tissues used in this study were frozen from the previous studies[32,44,51]. No animals were taken from the environment specifically for the present study.

Animals, whose tissues were used for phylogeographic analysis, were sampled by trawling in 2006, 2010, and 2015 in seven locations along the continental margin of the North Atlantic Ocean, covering a large range of temperature and salinity gradient (North America, Iceland, Norwegian Coast, Helgoland, Kattegat, Kiel Bay, White Sea; Table 1)[32,44,51]. For molecular, physiological, and ecological investigations based on specimens belonging to the Icelandic population, we used tissues from animals sampled in March 2010 in the northeast of Iceland (66°01.5 N 14°50.9 W). Before each experimentation, specimens were held in aquaria with recirculating water and course bottom sediment, to recover from sampling and transportation stress. Water parameters were adjusted to match the natural conditions at the collection site. Prior to dissection, animals were individually weighed, and the animal body weight was calculated by subtracting dry shell weight from the whole animal weight. Different tissue were sampled, frozen in liquid nitrogen, and stored at −80 °C[32,44,51]. Gonadic tissue was dehydrated, embedded in paraffin and histological sections were stained with haematoxylin/eosin solutions, observed under a Medicus HF: Hellfeld Microscope (Hund, Wetzlar, Germany) at 10x magnification to determine *A. islandica* sexes. Shell line of strongest growth (LSG) was measured with calipers to the nearest mm. Individual age (up to 226 years old) was deduced from shell growth bands as previously described[51].

**Molecular investigations**. For the phylogeographical study, total DNA was extracted from adductor muscle using the Qiagen DNeasy Blood & Tissue Kit, quantified using Nanodrop D-1000, and stored at −80 °C. Partial *cytochrome b* and *16 S* sequences were carried out using the same primers and conditions as previously described[32] (see Supplementary Data 2 for details) available sequenced amplicon: Supplementary_Data_3.fst & Supplementary_Data_4.fst).

To determine the mtDNA type of each individual (individuals carrying the ♀ or ♂ mtDNA), DNA from the adductor muscle sample was extracted using Chelex extraction. Briefly, a small piece of tissue was homogenized in 6% Chelex solution (50 mM Tris, 0.5 mM EDTA, pH 8.0) and agitated on vortex (2 times 30 s at maximum speed). The extract was centrifuged (2000 g, 2 min), incubated 10 min at 99 ˚C, and then centrifuged at 10,000 g for 10 min. The supernatant was sampled and amplified following multiplex PCR protocol: amplifications were carried out in 15 µL reaction volume comprising 1.5 µL 10X Taq buffer (5Prime), 0.3 µL dNTP mix (10 mM), 0.075 µL of each *cytochrome b* primer (100 µM), 0.075 µL of each 16 S primer (100 µM), 0.18 µL Hot Master Taq DNA polymerase (5U/µL, 5Prime, Hamburg, Germany) and 2 µL supernatant extract. Reactions were performed on Eppendorf MasterCycler (Supplementary Data 2) according to the manufacturer's instructions. The size of amplified products was checked on a 2% agarose gel in TBE buffer to determine if the individual carries the ♀ mtDNA (301 bp for *cytochrome b* and 377 bp for *16 S*) or the ♂ mtDNA (212 bp for *cytochrome b* and 503 bp for *16 S*). The sensitivity of this multiplex PCR was assessed on a mix of samples carrying only the ♂ or the ♀ mtDNA and showed that this technic allows the detection of both mtDNAs whatever their concentration (Supplementary Figure 5). To assess its reproducibility, the multiplex PCR was applied on 98 previously sequenced samples and perfectly succeed to detect the mtDNA type present.

The presence of each mitotype and their expression were assessed by quantitative PCR. Total DNA from the foot, gills, and mantle were extracted using the Qiagen DNeasy Blood & Tissue Kit, quantified using Nanodrop D-1000, and stored at −80 °C. Total RNA from gills, was isolated using the Qiagen RNeasy kit following the manufacturer's instructions. RNA was treated with RNase-free DNase set (1 h at room temperature; Qiagen), quantified using Nanodrop D-1000, and stored at −80 °C. Reverse transcription was realized using the MultiScribe Reverse Transcriptase kit following the manufacturer's instructions (Applied Biosystems). Partial ♀ and ♂ *cytochrome b* forms, *18 S*, *28 S*, and *β actin* presence or expression were investigated using HRM PCR Kit (Qiagen). Reactions were performed on Qiagen Rotor-Gene (Supplementary Data 2) according to the manufacturer's instructions. All samples, blank and negative control (DNA free total RNA before reverse transcription) were run in duplicate along with dilutions of known amounts of target sequence (standard curve) to quantify the initial DNA or cDNA concentration (Concentration = Efficiency$^{\Delta Ct}$). The results are expressed as the ratio of the target over the geometric mean of the best combination of two housekeeping genes determined using NormFinder software[52]. Melting curves were analyzed for all reactions, and we checked that only one amplicon was amplified during the qPCR and that for each couple of primer, the amplicon has the same characteristic among all samples and standard curve. Randomly chosen qPCR products were checked on a 2% agarose gel in TBE buffer and sequenced to control the quality and the accuracy of the amplification (see available sequenced amplicon Supplementary_Data_5.fst/Supplementary_Data_6.fst).

**Mitochondrial genome sequencing and analysis**. Total DNA was extracted from adductor muscle tissue of 18 specimens (16 carrying the ♂ mtDNA and 2 with the ♀ mtDNA) as previously described. The DNA was then converted to an Illumina sequencing library and each specimen barcoded. We obtained 12–15 million paired reads for each specimen. These data were assembled using abyss-pe and the mitochondrial sequences identified by searching for similarities with the previously published *A. islandica* mitochondrial genome via blast. The circular genomes were closed by sequencing a PCR product resulting from amplification of mitochondrial DNA using custom primers adjacent to the remaining gap.

All mitochondrial sequences were aligned within the Staden package to identify polymorphisms. MITOS online software[53] (revision 917) was used to delimit mitochondrial protein-coding genes, ribosomal and transfer RNA. Sequence differences (SNPs) between ♀ and ♂ mitochondrial genomes were counted in a sliding window of 500 with a step size of 1 and extracted using custom-made software tools. These differences were considered if, for one given position, nucleotides from all ♂ mitochondrial DNA sequences are different from nucleotides from all ♀ mitochondrial DNA sequences, considering gaps as a fifth nucleotide. MEGA 6[54] (v6.06) was used to calculate p-distance between sequences (with bootstrap analysis 1000 replicates). Mitochondrial genome sequences are available on GenBank (accession numbers MG838904 to MG838921). To these canonical ♀ and ♂ mtDNA sequences, the individual sequencing reads generated from the whole genome shotgun libraries were mapped using bowtie2[55]. Regions characteristic for either ♀ or ♂ mtDNA were extracted and visualized with integrated genome browser[56].

**Sequence analysis**. *Cytochrome b* and *16 S* sequences were aligned using the ClustalW algorithm of CodonCode Aligner program (v4.2.5, CodonCode Corporation, Dedham, MA, USA) and SeaView[57] (v4.5.4). Amino acid composition for *cytochrome b* was deduced using the invertebrate mtDNA genetic code (translation

Table 5). MEGA 6 (v6.06)[54] was used to calculate p-distance between sequences (with bootstrap analysis 1000 replicates). Maximum likelihood (PhyML) with bootstrap analysis (500 replicates) and Bayesian phylogenies (MrBayes) were performed on cytochrome b and *16 S* concatenated sequences with *Dosinia exoleta* (Bivalvia, Veneridae, Genbank accession number: *cytochrome b* - GQ166609.1; 16 S – JF808184.1) as outgroup (closest sequence according to BLAST search) using Hasegawa-Kishino-Yano, with Invariable site and discrete Gamma distribution, nucleotide substitution model as determined by jModelTest2[58] (BIC = 8905; lnL = 2351; version 1.6). The 127 bp indel occurring in the *16 S* partial sequences of individuals carrying the ♂ mtDNA was replaced by one nucleotide in this analysis and treated as a single event. FigTree[59] (v1.4.4) was used to edit the phylogenetic tree. DNA haplotype networks were realized using Haplotype Viewer[60].

Twelve proteins encoded by the mitochondrial genomes of Veneroida species available (including ♀ mtDNA and ♂ mtDNA consensus mtDNAs) were aligned using the PRANK web server (https://www.ebi.ac.uk/goldman-srv/webprank/)[61], ambiguously aligned columns in the resulting alignments were identified and removed using BMGE v. 1.12_1[62] on the ngphylogeny server (https://ngphylogeny.fr) and a concatenated alignment of 2697 amino acids was put together in Geneious Prime v2021.1.1. A Maximum Likelihood phylogenetic tree rooted with *Crassostrea* was calculated on the IQTree web server (http://iqtree.cibiv.univie.ac.at). The models of amino acid substitution best fitting to the data were estimated for each of the twelve partitions (representing protein-coding genes) separately. The tree topology was tested for robustness by calculating the ultrafast bootstrap support in IQTree and the tree was edited in ITOL (https://itol.embl.de).

**Biochemical investigations**. Each gill sample was weighed and homogenized in 6 volumes of cold homogenization buffer (20 mM Tris (hydroxymethyl) amino-methan, 1 mM EDTA, 0.1% Tween 20, pH 7.4) using a Precellys Homogenisator 24 (Bertin Technologies) with 2 times 15 s at 5000 rpm at 8 ˚C. Enzyme activities were measured at 8 ˚C using a Plate Reader TriStar (Berthold Technologies) and expressed in U per gram of tissue. All assays were run in duplicate. All protocols were adapted from Breton and collaborators[29].

**Statistics and reproducibility**. All statistical tests were realized using R 3.2.1[63]. The effect of sex and mitotype on enzyme activities was analyzed using Kruskal Wallis following by the Wilcoxon Post hoc test. A between-groups principal component analysis (PCA) was used to assess the variation in morphological traits and enzymatic activities in the Icelandic population. This analysis consists to running PCA on a dataset where observations are gathered by user-defined groups to emphasize the between-groups variability in the analysis process[64]. Here groups were built by distinguishing specimens carrying the ♀ and ♂ mtDNA in females and males of *A. islandica* populations. The first two axes contributed to 68.9% and 30.4% respectively of the explained variance. The significance of the proportion of the variability explained by the grouping factor (e.g. mitotypes and sex) in between-groups PCA was tested by Monte-Carlo permutation (999 permutations). Between-groups PCA was performed using 'ade4' library[65]. The effect of age and mtDNA type were analyzed on length of strongest growth by 3 parameters asymptotic exponential model ($y = a − be^{−cx}$) and on animal weight and shell weight by Michaelis Menten model ($y = ax/(1+bx)$).

**Reporting summary**. Further information on research design is available in the Nature Research Reporting Summary linked to this article.

## Data availability
Mitochondrial genome sequences are available on GenBank (accession numbers MG838904 to MG838921). Other data that support the findings of this study are available from the corresponding author upon request.

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

## Acknowledgements

We thank Dr. Eva Philipp for providing *Arctica islandica* samples from the Icelandic population for the molecular, physiological and ecological investigations. We particularly thank Andrea Eschbach and Stephanie Meyer for their technical support. The work was financially supported by the Alexander von Humboldt foundation (http://www.humboldt-foundation.de) and the Auvergne Rhone Alpes region to C.D.

## Author contributions

C.D., D.A., and C.H. designed all experiments and conducted animal samplings. H.G. realized sampling in Iceland, measured and aged the animals. C.D. performed all experiments with help of G.G. for the whole genome sequencing. C.D., D.A., G.G., B.A., and C.H. analyzed the data. C.D. and B.A. performed the statistical analysis. C.D., D.A., G.G., B.A., and C.H. wrote the paper.

## Funding

## Competing interests

The authors declare no competing interests.
