## [Transparent Peer Review File · Communications Biology]

Reviewers' comments:

Reviewer #1 (Remarks to the Author):

The Manuscript entitled "Exclusive presence of male mitochondria in somatic tissues impacts bivalves mitochondrial functioning" by Cyril Dégletagne, Doris Abele, Gernot Glöckner, Benjamin Alric, Heike Gruber, and Christoph Held presents new and interesting data about the distribution of male-inherited mitochondria in somatic tissues of *Arctica islandica*, a bivalve species showing Doubly Uniparental Inheritance (DUI).

The work includes: new full length M-type mtDNAs; analysis of the occurrence of M-type mtDNA in populations from Norway, Iceland, Baltic Sea, Helgoland, US, Kattegat and the White Sea; the finding of male and female individuals with predominance ("exclusive presence") of M-type mtDNA in the investigated somatic tissues; distribution of F- and M-type mt lineages in somatic tissues; analysis of the transcription of F- and M-type mtDNA across tissues and sexes; F- and M-type mtDNA activity essays; analysis of the correlation between growth rates and mt type in soma.

The main findings can be resumed as follows:

- 1) *A. islandica* F- and M-type mtDNAs differ by ~5.55 at the DNA sequence level. It is a low divergence compared to that of other DUI species, and indicates that the two lineages splitted more recently.
- 2) The features of *A. islandica* mtDNA are similar to that of other bivalves (especially Veneridae). No *cox2* duplication or extension has been found.
- 3) M-type mitochondria is detected in somatic tissues but only in subarctic populations;
- 4) Some males and females seem to have only the M-type in their somatic tissues.
- 5) Enzymatic essays show a reduced activity in somatic tissues showing only M-type mtDNA;
- 6) The presence of M-type mitochondria in somatic tissues might affect growth rates.

The Doubly Uniparental Inheritance of mitochondria (DUI) has been my main research topic since my second year of PhD (2007). Overall, I think this is a very interesting work, showing exciting and intriguing observations that certainly add to the knowledge of the DUI community and will shape future research.

The work is methodologically sound (with just a minor concern about samples) and all the analyses seem appropriate to me. The narrative is well-written and clear. Some recent bibliography is missing (BUT see important note below).

Sometimes I think the Authors should be less sweeping and more cautious about the conclusions they are drawing from their data (maybe even the title might be more... prudent). While I think it is important to advance working hypotheses and even speculate a bit, I was uncomfortable with some statements the Authors made.

See my specific comments below.

IMPORTANT NOTE: throughout the review I cited a few papers, some of which authored also by me. I AM NOT asking the Authors to include such references in a reviewed version of the manuscript. I added such references only to support my views, or to provide examples. It is up to the Authors to decide which references they want to include, if any.

#SPECIFIC COMMENTS

> Lines 22-23

"Metazoans normally possess a single lineage of mitochondria inherited from the mother (♀-type mitochondria) whilst ♂-type mitochondria are typically eliminated."

In species with strictly maternal inheritance there is only one mt "type" (namely, there are no divergent mt lineages), so I think referring to paternal mtDNA as "♂-type" in non-DUI species can be confusing.

> Lines 23-25

"In doubly uniparental inheritance (DUI), σ -type mitochondria are retained in male gonads of certain bivalve species and, in few of these, limited amounts of σ -type mitochondria co-exist with ♀ -type in somatic tissues (paternal leakage)."

The term "paternal leakage" in a mitochondrial context is mainly used to indicate the inheritance of paternally-derived mtDNA through generations. In this case it indicates the presence of a lineage of mtDNA which is normally inherited through sperm (and supposedly limited to gonadal tissue, more of this below) in somatic tissues, which is a different thing. In my experience, most of the people not familiar with the DUI system, tend to confuse an evolutionarily stable pattern of paternal mt inheritance (like that occurring in DUI species) with "leakage" (in the sense of accidental, instable or occasional transmission of a genetic variant outside the normal "route"). Therefore, even if I get what the Authors meant with that sentence, I would avoid the use of "paternal leakage" in such acception.

> Line 27

"A. islandica."  *Arctica islandica*

> Lines 41-43

"The functioning of mitochondria is finely tuned to the cellular and nuclear background of their "host" cell, which places the gene products encoded in the mitochondrial DNA (mtDNA) sequence under tight selection to remain compatible with their nuclear encoded interaction partners (mito-nuclear coevolution)."

I think this description about mito-nuclear coevolution is concise, but it has a "finalistic" vibe that makes me uncomfortable. Please consider rephrasing.

> Line 46

"and nearly neutral variants derived from it"

I am not sure what the Authors imply here. Why specify "nearly neutral"? mtDNA is also subject to selection: purifying selection is quite evident from molecular analyses across eukaryotes, but increasing evidence of adaptive evolution is also accumulating. So I do not understand the need of limiting the derived variants to the "nearly neutral" ones.

> Lines 59-60

"in a growing number of dioecious bivalve species belonging to the orders Veneroida, Unionida and Mytilida"

Please add Nuculanoida (see Gusman et al. 2016, reference #18)

> Line 62

"is restricted to gonochoric bivalves"

The best citation for this would be Breton et al. 2011 (see below).

Interestingly, Artur Burzynski and Collaborators recently reported an hermaphroditic mussel with DUI (Lubośny et al, 2020, see below). So it looks like we have to update our view about such DUI feature.

Refs:

- Breton S, Stewart DT, Shepardson S, Trdan RJ, Bogan AE, Chapman EG, Ruminas AJ, Piontkivska H, Hoeh WR. 2011. Novel protein genes in animal mtDNA: a new sex determination system in freshwater mussels (Bivalvia: Unionoida)? *Mol. Biol. Evol.* 28:1645–1659.

- Lubośny M, Przyłucka A, Śmietanka B, Burzyński A. 2020. *Semimytilus algosus*: first known hermaphroditic mussel with doubly uniparental inheritance of mitochondrial DNA. *Sci. Rep.* 10:11256.

> Lines 63-65

"During early embryogenesis of DUI species, the σ -type mitochondria are recognized and then eliminated in females. In males, however, σ -type mitochondria persist and are aggregated in one blastomere destined to produce the male germline (i.e. gonad)."

This is the scheme of the DUI mechanism that has been used since the early days (1990s). New observations indicate that such mechanism should be revised. In a recent publication, in *Ruditapes philippinarum* we observed heteroplasmy at the organelle level in undifferentiated germ cells of both sexes and in male soma, whereas gametes were homoplasmic: eggs for the F-type and

sperm for the M-type (Ghiselli et al. 2019, see below).

Ref:

- Ghiselli F, Maurizii MG, Reunov A, Ariño-Bassols H, Cifaldi C, Pecci A, Alexandrova Y, Bettini S, Passamonti M, Franceschini V, et al. 2019. Natural Heteroplasmy and Mitochondrial Inheritance in Bivalve Molluscs. *Integr. Comp. Biol.* 59:1016–1032.

> Lines 65-71

"This leads to the formation of heteroplasmic males with ♀-type mitochondria in all somatic tissues and ♂-type mitochondria in their gonads and sperm. DUI females are usually homoplasmic with only ♀-type mitochondria in somatic and gonadic cells. Even in DUI species, somatic cells of male and females are strictly homoplasmic with only ♀-type mitochondria present. Malfunctions in segregation and/or incomplete elimination of ♂-type mitochondria during early embryogenesis cause heteroplasmy in somatic tissues of single individuals, both male and female, in a small subset of DUI species."

This section contains too much generalization in my opinion. The presence of M-type in male somatic tissues is known to occur in *R. philippinarum* (Ghiselli et al. 2011, reference #22), *Venustaconcha ellipsiformis* and *Utterbackia peninsularis* (Breton et al. 2017, reference #14), and also in *Mytilus galloprovincialis* (see for example Obata et al. 2006 below). The same works reported also that M-F heteroplasmy is pretty common. The Authors are reporting a simplified scheme of the DUI system, as the most of us working on DUI have done for years. That said, evidence against that scheme is accumulating fast and in my opinion is now the time to acknowledge some deviations from the simplified mechanism reported here. In particular, the statements "DUI females are usually homoplasmic with only ♀-type mitochondria in somatic and gonadic cells" and "Even in DUI species, somatic cells of male and females are strictly homoplasmic with only ♀-type mitochondria present" we now know that are not true.

In my opinion, the last sentence ("Malfunctions in segregation and/or incomplete elimination of ♂-type mitochondria during early embryogenesis cause heteroplasmy in somatic tissues of single individuals, both male and female, in a small subset of DUI species") is not sufficient to compensate for the generalization discussed above, because according to it, the observations deviating from the simple version of the mechanism are due to malfunction and are limited to "a small subset of species". I disagree with this view: we do not know which is the "rule" and which is the "exception", and since most of the DUI species have not been investigated thoroughly for the M/F-type distribution in different tissues, I would not say that the observed deviations pertain to only a small subset of species. We need to investigate a broader range of taxa to get a glimpse of the conserved features of the DUI mechanism, that is why this work is so important.

Ref:

Obata M, Kamiya C, Kawamura K, Komaru A. 2006. Sperm mitochondrial DNA transmission to both male and female offspring in the blue mussel *Mytilus galloprovincialis*. *Dev. Growth Differ.* 48:253–261.

> Lines 119-120

"The sites that distinguish the two mitochondria"

Divergent sites?

> Lines 121, 123 and on

Gene/protein abbreviations: please use lower case italics for genes and upper case for proteins.

> Line 123

Delete "at the DNA level".

> Lines 126-128

"Although DUI species are frequently characterized by a 3' extension or duplication of their COX2 gene, a feature that is believed to be involved in the transmission of ♂-type mitochondria"

 "Although DUI species are frequently characterized by a 3' extension or duplication of their COX2 gene, a feature that some Author suggested to be involved in the transmission of ♂-type mitochondria"

> Figure 2

Not a mandatory request at all, but as a fan of big maps of mt genomes, I would really like a larger (and clearer) map of M-and F-type mtDNA in *A. islandica* (such those obtained with Circos, OGDRAW, etc...).

> Lines 140-141

"we found that 300 male and female individuals from all sampled populations had the typical metazoan configuration with only ♀-type mitochondria in somatic cells (adductor muscle)."

Just a note: it is possible that some of such "typical" individuals carry M-type mtDNA in other somatic tissues (mantle, gill, digestive gland, foot...). I would not consider the adductor muscle as a proxy for all the somatic cells (see below)

> Lines 141-142

"In 15 individuals, however, we detected only ♂-type mitochondrial DNA in somatic tissues."
See above.

> Lines 175-179

"In other DUI species the ♂-type mitochondria, typically confined to the male gonads, are absent in somatic cells and, hence, do not occur in females at all. Cases of ♂-type mitochondrial "leakage", characterized by the co-presence of ♂- and ♀-type mitochondria in somatic tissue, have been reported for some DUI species, mostly in male but also female individuals"

These first statement is not true (please see comments for Lines 65-71) and the second is inaccurate (please see comments for Lines 23-25).

> Lines 193-195

"These results, in combination with the complete absence of ♀-type mitochondria with predominant ♂-type mitochondria in foot and adductor muscle (Fig. 4), rule out paternal leakage as an explanation of the observed pattern."

Two notes: 1) I am uncomfortable with statements of "complete absence" in a scientific paper; 2) I did not get the last part ("rule out paternal leakage as an explanation of the observed pattern").

> Line 218

What does "ng" stands for? I would expect a copy number or a ratio of some sort (e.g. with respect to housekeeping genes).

> Line 223

Given the experimental design I would refer to "transcription" rather than "expression". It's a long way from RNA to final product, especially in mitochondria.

> Line 224

"The number of active mitochondria per cell is linked to the cellular metabolic requirements of the organism and can be roughly estimated by the mtDNA copy number"

The mtDNA may roughly estimate the number of mitochondria (even if I am less and less convinced about this), but in my opinion cannot estimate the number of ACTIVE mitochondria. An inactive mitochondrion still has mtDNA (which is included in the copy number), but it does not have physiological activity... So I think the term "active" should be avoided.

> Lines 235-265

Recently, Sophie Breton and Collaborators published two very interesting papers about mitochondrial activity in DUI bivalves. Since these kind of studies are still quite rare in bivalves (especially those with a DUI focus), I think it would be very interesting if the Authors would comment their results in the light of those of Bettinazzi et al (2019, 2020, see below).

Refs:

- Bettinazzi S, Nadarajah S, Dalpé A, Milani L, Blier PU, Breton S. 2020. Linking paternally inherited mtDNA variants and sperm performance. *Philos. Trans. R. Soc. Lond. B Biol. Sci.* 375:20190177.

- Bettinazzi S, Rodríguez E, Milani L, Blier PU, Breton S. 2019. Metabolic remodelling associated with mtDNA: insights into the adaptive value of doubly uniparental inheritance of mitochondria. *Proc. Biol. Sci.* 286:20182708.

> Lines 266-275

I think this interpretation of the results is very interesting and indicates potential future investigations. That said, in my opinion the concepts should be expressed with a more dubitative style. The scenario hypothesized by the Authors is certainly possible, but I think it should be presented as a working hypothesis, not an established fact.

> Line 278

"The fact that these genuine underperformers"

I would not say that yet. The biochemical activity of M-type mitochondria might just be different, not worse.

> Lines 277-305

This part is very interesting as well, and presents some intriguing questions. I am not yet convinced about the existence of a causal correlation between mt type and growth, because I can see the correlation, but of course a causal link needs further study. One of my concerns is that since mt type and sex are linked in most cases, I wonder if the observed correlation is not between mt type and growth, but between sex and growth. Many studies in bivalves have linked sex and growth, and in *R. philippinarum* we observed that females are usually bigger (heavier) than males.

That said, I appreciate the cautious tone the Authors used here, so I do not see any problem with some healthy speculation.

> Lines 322-323

"Here we describe the first clear example in metazoans with a complete replacement of ♀-type by ♂-type mitochondria in somatic tissue."

Again, this sounds a tad weird to me because M-type mtDNA (namely a mitochondrial lineage inherited through sperm and well separated by the egg-transmitted mitochondrial lineage) does not exist in species with strictly maternal inheritance. So in my opinion this sentence is technically true, but confusing.

> Line 325

"The exclusive presence of ♂-type mitochondria in somatic cells, similar to the paternally inherited mtDNA in sperm might be caused by failure of the DUI specific selection, feminization of ♂-type mitochondria, or by the strict paternal inheritance of ♂-type mitochondria."

I find this sentence confusing. In particular:

"similar to the paternally inherited mtDNA in sperm"

The M-type in somatic tissues comes from the mtDNA carried by sperm. In what sense "similar"?

"feminization of ♂-type mitochondria"

Possible, but as far as I know there is no evidence yet of such a thing. As things stand now, it is not a parsimonious hypothesis.

"or by the strict paternal inheritance of ♂-type mitochondria"

I do not understand. M-type is strictly paternally inherited in DUI species...

> Lines 325-327

"This unprecedented inheritance system places mitochondria in novel nuclear backgrounds and might afford adaptive advantages in novel or changing environments."

I disagree about this being an unprecedented inheritance system. The inheritance system is the DUI system, it is the amount of M-type in somatic tissues that is interestingly different and, in the case of individuals carrying mostly/only M-type mtDNA, unprecedented.

> Lines 348, 351, 353-355

The Authors used frozen tissues sampled in March 2010. This might originate some concerns. If properly stored, frozen tissues are still good for DNA extraction (even if, in my experience, the older they get, the more difficult is to get good DNA), so the DNA analyses should be fine. I have more concerns about RNA. In my experience it is really difficult to get good quality RNA from old samples. In this case, the approach is a RT-qPCR, so it is possible that the amplification of small fragments of RNA is unaffected by degradation, but I would take the results with a grain of salt.

I am not an expert about mitochondrial activity essays so I don't know the impact that old samples could have had on the results.

All that considered, I do not think that this issue undermines the work which I think it is solid, but I would recommend to be cautious when presenting the results and drawing conclusions.

> Line 456

"A between-groups principal analysis"

 "A between-groups principal component analysis"

#END

Reviewer #2 (Remarks to the Author):

I reviewed this manuscript already and I rejected it. I am very disappointed to see that the authors did not change a single word. So do I... The data (and analyses) are not robust and lack of novelty. Because of that, I (again) recommend rejection of the paper. Below are my main concerns.

1. The "DUI aspect" of the paper

The authors claim that the two divergent mitotypes they sequenced in *A. islandica* represent the paternally- and maternally-derived mtDNAs typical of DUI species. The authors obtained their sequences from somatic tissues of different individuals from different populations. What surprises me is that they speculate on the fact that they represent DUI lineages when they provide no evidence at all that these two divergent mitotypes are associated with sex and gametes. Based on the results presented here, there is no indication that the two genomes are respectively paternally- and maternally-derived, actually neither enough evidence to claim that this is a species with DUI.

The main feature of a DUI system is that two mitochondrial lineages are strictly associated with anisogamous gametes, they coexist in the same populations and are species-specific. To claim that *A. islandica* is a DUI species, the sperm in ALL populations should bear the haplotype that the authors claim to be a paternally-derived mtDNA. The fact that this putative paternal haplotype is absent in almost all populations tested suggests that this is probably not the case.

In their results and discussion section the authors wrote that they have previously successfully amplified the so-called σ -type mtDNA in sperm of the same species (Dégletagne et al., 2016). However, this actually never happened. They did not check sperm, but male gonads, which, if we consider *A. islandica* as a DUI species, contain both haplotypes (gonads also contain somatic cells). There is no evidence at all that this "putative paternal mitotype" is transmitted by father to sons and, as such, there is no way to pretend it to be a paternally-derived lineage. Moreover, as males are undoubtedly present in all populations tested, so should be the here-called paternally-transmitted mtDNA.

The existence/persistence of two sex-linked mitochondrial lineages is species-specific and cannot be population-specific for a DUI species. No matter the location of various populations, the genetic pool of e.g. *M. edulis* (and other DUI species) always contains both sex-linked mtDNA variants. In this paper, the so-called paternal haplotype was only found in 15 out of 315 individuals and restricted to somatic tissues of one (or two) specific population. How the authors would explain the existence of DUI in just one population (and even there, just 11% of the individuals bear the divergent mtDNA)? A DUI system is species-specific and never population-specific. Also, somatic tissues in DUI species, when they contain the paternally-derived mtDNA, are always heteroplasmic, and this is not the case here. This means that either the "divergent haplotype" found is maternally-inherited or that the methods used to detect both "putative DUI mitotypes" do not work. According to figure S5, the "putative paternal haplotype" has to be in low concentration to be able to detect the "maternal haplotype" using the multiplex PCR, so if it is in relatively high concentration, a heteroplasmic state cannot be detected. Also, the figure 3 is not really convincing, there is no detail about the "sex-specific regions" chosen – why not testing the entire mtDNAs? (actually this is what the authors wrote in the results section but according to the figure, this is not true).

The authors proposed that the absence of the paternally-derived in some populations could be the

result of a strong selection against this "variant" in individuals living in areas characterized by a more variable environmental condition. In my opinion, it is more likely the opposite, as heteroplasmy could foster within population phenotypic plasticity (in this case, at least, as revealed by the OXPHOS effect), which could be better exploited in variable environments than in stable ones. In addition, the populations the authors studied are not so distant and share a pretty common environment. Even if the author proposed the environment to be very different, and I can agree that some differences surely exist, they did not compare populations living in different hemispheres, or Caribbean vs Arctic seas, but populations from Norway, Iceland, Baltic Sea, Helgoland, Kattegat, White sea etc. (all these individuals live in cold seas of the North hemisphere and they all live burrowed in sand at the bottom of the sea). For example, *R. philippinarum* (a DUI species) possesses the two "typic DUI lineages" no matter which population you test, and we are talking about populations of the Philippines, Italy, west coast of North America etc. The two mitotypes found in the present study might just represent two regional mitochondrial variants present within the same species, one widespread among the *A. islandica* populations and one underrepresented, specific of few individuals in the populations around Iceland.

Overall, if *A. islandica* is a species with DUI, and if the two detected mitotypes are really the maternally- and paternally-derived mt lineages, they should be present in all the populations (and be strictly associated with sex and gametes), and this is not the case here (or has not been correctly tested). The two mitotypes described could only reflect within species and between population mitochondrial heterogeneity found in somatic tissues of *A. islandica* individuals independently of sex (found in both females and males), and not associated with gametes. This cannot in any way be confused with the species-, sex- and gamete-specific association of all the DUI lineages described so far (in more than 100 species). As such, this paper is highly speculative (or incorrect).

Also, several complete maternal and paternal mt genomes of DUI species are present in GenBank and they all have something in common: they possess additional sex-specific protein-coding genes. Did the authors verify for the presence of such genes? The two mitotypes provided here are not so dissimilar (5.5% of divergence or less) compared to the standard DUI species divergence (from 20 up to 50%).

2. The "mitochondrial functional aspect" of the paper

The authors claim that the expression of σ -type mtDNA in somatic tissues reduces mitochondrial activity. Although interesting per se, the results presented here are in line with what has already been widely seen in several DUI species and do not provide anything new to the community other than supporting previous evidence. The phenotypic effect of bearing sex-linked mitochondria in DUI species has already been observed several times during the last decades. Starting from the effect of bearing male or female derived mitochondria upon sperm motility and bioenergetic strategy to the effect upon gametes and somatic tissue energy metabolism (Everett et al., 2004; Jha et al., 2008; Stewart et al., 2015; Bettinazzi et al., 2020). Concerning the OXPHOS, the activity of the two sex-linked mt lineages has been investigated (Milani and Ghiselli, 2015; Ghiselli et al., 2018) and even the depression in CCO activity was described ten years ago in *M. edulis* when comparing sperm carrying either the paternally-derived or the maternally-derived DUI haplotype (classic vs masculinized) (Breton et al., 2009). Finally, in *M. edulis* and *A. islandica* (interestingly, the very same species of this research) an OXPHOS remodelling of M-type DUI mitochondria was recently described in both gametes and somatic tissues (Bettinazzi et al., 2019). Concerning the experimental set up, it is unclear if the gills have been frozen before being weighted for the enzymatic analyses (according to the first paragraph of the methods no), so the correction for mg of tissue is probably not appropriate and activities should be corrected for mg of proteins instead.

Also, the authors claim that the metabolic changes presented here can influence female growth. To sustain such hypothesis, you need to 1) provide a viable explanation on how this could be possible and 2) base yourself on more data. In my opinion, you need to provide a more exhaustive vision of cellular bioenergetics, not limited to ETS and CCO enzymes. A decreased activity of ETS (CI+CIII) and CCO is already known in DUI species and it would have been interesting having clues about the activity of other enzymes related with the energy metabolism. This includes enzymes forming part of the OXPHOS itself, as well as glycolysis, cytosolic fermentation pathways, fatty acid metabolism etc. It would have been interesting to test a panoply of enzymes, not only the two for which extensive evidence exists in the literature, in order to have a bigger vision of the effect upon cell energy metabolism. For example, concerning the OXPHOS it would have been nice

to test CII. Since it is totally encoded by the nucleus, its activity would provide clues on whether the metabolic depression seen in CCO is really due by the interaction between a divergent mitotype and the common nuclear background.

Finally, the authors are proposing that the replacement of one haplotype by the other in females might be evolutionary relevant and influence females *A. islandica* growth rates. Specifically, the authors claim: "Curiously, we found that females with somatic σ -type mitochondria tend to have higher standardized residuals for all parameters tested, which might indicate that they are heavier and have a bigger and heavier shell than those with ♀ -type somatic mitochondria."

I found this a clear overstatement. First, if evolutionary relevant, I would expect this condition to be more present at both within- and between-population levels. This putative replacement was not revealed in most population, and, when revealed, only 11% of the individuals was carrying it.

Overall, saying that bearing a specific haplotype might be beneficial for females is highly speculative because:

- only 6 females carry the condition out of 315 total individuals;
- the authors base their claim on morphological parameters results which are not statistically significant (figure s4) and, even if they were, the $n=6$ is actually very small
- How could the authors claim that the changes in activities of ETS and CCO has a female-specific effect on growth when their enzymatic measurements was not accounted for males and females separately (figure 5b,c)? i.e. the activities were calculated for all the sexes mixed together, not like citrate synthase (figure 5d). As such we cannot know whether the expressed mitochondrial phenotype might change among sexes (if there is a sex-specific phenotypic effect of bearing the specific haplotype).
- How a lower activity of two ETS enzymes could explain a higher growth rate?
- Moreover, given that both males and females appear to bear this putative paternal haplotype in this population, why only females (and within females, only 6 of them) would be subjected to that?

Overall, what is the metabolic property that allows only few female individuals to grow faster and why this haplotype is not widespread among populations?

Also, the authors claim that " σ -type mitochondria, while not presenting an explicit advantage, are better tolerated at the high latitude by subarctic populations." I am confused. For the authors, does this specific haplotype confer an advantage or not?

Moreover, the authors state: "Assuming this σ -type mitotype is associated with a reduced ROS production, linked with the decrease of mitochondrial complex activities (Fig. 5B- 5C) in the mitochondria, this might even support extreme longevity in *A. islandica*."

- How did the authors assume a reduced ROS production in this species if they did not measure neither ROS production nor the antioxidant activity?

- Where the individuals with complete replacement older than the others 300? Is there any indication that they would have lived longer? No.

On the contrary, in DUI species, the paternal-mitochondria activity appears to link with an increased ROS production (Monagan and Metcalfe, 2020; Milani and Ghiselli et al., 2015; Bettinazzi et al., 2019a). Also, as proposed by some studies on *A. islandica* (Munro et al. 2013; Blier et al., 2017; Rodriguez et al., 2019), the "stoichiometric reorganization of OXPHOS components in *A. islandica* that might account for its low ROS production entail increasing the proportion of components upstream and downstream of the major ROS producing complexes (which are CI+III), that would allow reducing H₂O₂ efflux while maintaining total catalytic capacity." By a fast look at figure 5 the stoichiometry between enzymatic complexes appears to be conserved, and not in line with a potential decreased ROS production associated with a putative paternal haplotype. Since there is no measure of ROS production (or evidence of changes in antioxidant activity), forcing a link between longevity in this species and the activity of just two enzymes is, in my opinion, completely speculative.

Reviewer #3 (Remarks to the Author):

Studies of DUI are especially interesting when they can provide new insights into strictly maternal inheritance itself. This study is useful for both the field of DUI and the study of mitochondrial inheritance by providing potential consequences of the maintenance (instead of removal) of male-type mitotypes. Although the study relies on a few exceptional individuals of the study species

(which the authors have clearly stated), it will provide another direction for future studies of bivalves possessing DUI. Based on the methods provided I believe this type of study is repeatable with other bivalve populations.

With the above remarks in mind, I do have some specific questions and comments on the manuscript. Some larger questions I have:

(1) Were novel f- and m-orfs identified in individuals with complete mtDNA sequencing ? To my understanding venerid species do not necessarily possess the f- and m-orf which are of interest to studies of mytilid and unionid DUI species.

(2) Why was atp8 excluded from Figure 2 ? In other DUI possessing bivalves it can be difficult to identify, was this the case here ?

(3) in regards to line 416: Are these SNPs or SNVs ? Are the variations represented in at least 1% of the population to qualify as a SNP ? If so, is this supported by the paper ?

I believe a recent paper on two other venerid species would be of interest to this study (Capt et al. 2020 <https://doi.org/10.1038/s41598-020-57975-y>). It presents additional sequences that could supplement their tree (Figure 1) by introducing examples of DUI in two additional families (Solenioidea and Tellinoidea). It provides an interesting example of the cox2 extension the authors have discussed. Additionally, the findings may contribute to the discussion of divergence in lines 104-108.

Additionally, I have some specific comments:

Line 60: DUI is also found in Nuculanida (<https://doi.org/10.1007/s00227-012-2099-y> and [10.7717/peerj.2760](https://doi.org/10.7717/peerj.2760)).

Line 60-61: It is not yet clear what percent divergence between mitotypes in DUI is "typical", but there has been "lows" of 20% and "highs" of 50% documented. A reported range might be more clear to the reader, which the authors have done later (line 108).

Line 59/62: The authors use "dioecious" to describe separate sexes in a species, but later use "gonochoristic", using one term consistently may benefit readers new to DUI research. Although Fig. S2 and Fig. S3 are supplementary I believe they would benefit from the IDs being cleaned up for legibility or else locations be replaced by a colour legend, etc.

Fig. S5. Would benefit from the ladder bands being labeled with their size, even though the products are labeled it would assist those who wish to repeat these methods.

Dear reviewers,

Thank you for your comments on our paper. We respectfully and thankfully acknowledge the insightful and constructive criticism on our paper, and have done our best to correct and improve the critical parts of the manuscript.

Cyril Dégletagne, in the name of the entire authoring team.

Reviewers' comments:

Reviewer #1 (Remarks to the Author):

The Manuscript entitled "Exclusive presence of male mitochondria in somatic tissues impacts bivalves mitochondrial functioning" by Cyril Dégletagne, Doris Abele, Gernot Glöckner, Benjamin Alric, Heike Gruber, and Christoph Held presents new and interesting data about the distribution of male-inherited mitochondria in somatic tissues of *Arctica islandica*, a bivalve species showing Doubly Uniparental Inheritance (DUI).

The work includes: new full length M-type mtDNAs; analysis of the occurrence of M-type mtDNA in populations from Norway, Iceland, Baltic Sea, Helgoland, US, Kattegat and the White Sea; the finding of male and female individuals with predominance ("exclusive presence") of M-type mtDNA in the investigated somatic tissues; distribution of F- and M-type mt lineages in somatic tissues; analysis of the transcription of F- and M-type mtDNA across tissues and sexes; F- and M-type mtDNA activity essays; analysis of the correlation between growth rates and mt type in soma.

The main findings can be resumed as follows:

- 1) *A. islandica* F- and M-type mtDNAs differ by ~5.55 at the DNA sequence level. It is a low divergence compared to that of other DUI species, and indicates that the two lineages splitted more recently.
- 2) The features of *A. islandica* mtDNA are similar to that of other bivalves (especially Veneridae). No *cox2* duplication or extension has been found.
- 3) M-type mitochondria is detected in somatic tissues but only in subarctic populations;
- 4) Some males and females seem to have only the M-type in their somatic tissues.
- 5) Enzymatic essays show a reduced activity in somatic tissues showing only M-type mtDNA;
- 6) The presence of M-type mitochondria in somatic tissues might affect growth rates.

The Doubly Uniparental Inheritance of mitochondria (DUI) has been my main research topic since my second year of PhD (2007). Overall, I think this is a very interesting work, showing exciting and intriguing observations that certainly add to the knowledge of the DUI community and will shape future research.

The work is methodologically sound (with just a minor concern about samples) and all the analyses seem appropriate to me. The narrative is well-written and clear. Some recent bibliography is missing (BUT see important note below).

Sometimes I think the Authors should be less sweeping and more cautious about the conclusions

they are drawing from their data (maybe even the title might be more... prudent). While I think it is important to advance working hypotheses and even speculate a bit, I was uncomfortable with some statements the Authors made.

See my specific comments below.

IMPORTANT NOTE: throughout the review I cited a few papers, some of which authored also by me. I AM NOT asking the Authors to include such references in a reviewed version of the manuscript. I added such references only to support my views, or to provide examples. It is up to the Authors to decide which references they want to include, if any.

With my best Regards,

Fabrizio Ghiselli, PhD

Answer: we changed the title and re wrote entire paragraphs of the paper.

#SPECIFIC COMMENTS

> Lines 22-23

"Metazoans normally possess a single lineage of mitochondria inherited from the mother (♀-type mitochondria) whilst ♂-type mitochondria are typically eliminated."

In species with strictly maternal inheritance there is only one mt "type" (namely, there are no divergent mt lineages), so I think referring to paternal mtDNA as "♂-type" in non-DUI species can be confusing. I have exchanged ♂-type by paternal mtDNA... that makes sense for non DUI species which logically have no "male type!!"

Answer: We changed it accordingly. Our original decision to use ♂-type instead of paternal mtDNA was to emphasize that mtDNA can be independent of the sex of the individual bearing the mitochondria (we report females having male-type mitochondria in somatic cells) but realize that this can be confusing.

> Lines 23-25

"In doubly uniparental inheritance (DUI), ♂-type mitochondria are retained in male gonads of certain bivalve species and, in few of these, limited amounts of ♂-type mitochondria co-exist with ♀-type in somatic tissues (paternal leakage)."

The term "paternal leakage" in a mitochondrial context is mainly used to indicate the inheritance of paternally-derived mtDNA through generations. In this case it indicates the presence of a lineage of mtDNA which is normally inherited through sperm (and supposedly limited to gonadal tissue, more of this below) in somatic tissues, which is a different thing. In my experience, most of the people not familiar with the DUI system, tend to confuse an evolutionarily stable pattern of paternal mt inheritance (like that occurring in DUI species) with "leakage" (in the sense of accidental, instable or occasional transmission of a genetic variant outside the normal "route"). Therefore, even if I get what

the Authors meant with that sentence, I would avoid the use of "paternal leakage" in such acceptance.
Answer: We deleted the term (see above)

> Line 27

"A. islandica."  Arctica islandica

Answer: We changed it

> Lines 41-43

"The functioning of mitochondria is finely tuned to the cellular and nuclear background of their "host" cell, which places the gene products encoded in the mitochondrial DNA (mtDNA) sequence under tight selection to remain compatible with their nuclear encoded interaction partners (mito-nuclear coevolution)."

I think this description about mito-nuclear coevolution is concise, but it has a "finalistic" vibe that makes me uncomfortable. Please consider rephrasing.

Answer: Thank you, we rephrased : The genes encoded by the mitochondrial DNA (mtDNA) sequence are finely tuned to the cellular and nuclear background of their "host" cell, which supports and conserves mitonuclear interactions and mitochondrial biology.

> Line 46

"and nearly neutral variants derived from it"

I am not sure what the Authors imply here. Why specify "nearly neutral"? mtDNA is also subject to selection: purifying selection is quite evident from molecular analyses across eukaryotes, but increasing evidence of adaptive evolution is also accumulating. So I do not understand the need of limiting the derived variants to the "nearly neutral" ones.

Answer: Agreed, we deleted this part

> Lines 59-60

"in a growing number of dioecious bivalve species belonging to the orders Veneroida, Unionida and Mytilida"

Please add Nuculanoida (see Gusman et al. 2016, reference #18)

Answer: We added the reference

> Line 62

"is restricted to gonochoric bivalves"

The best citation for this would be Breton et al. 2011 (see below).

Interestingly, Artur Burzynski and Collaborators recently reported an hermaphroditic mussel with DUI (Lubośny et al, 2020, see below). So it looks like we have to update our view about such DUI feature.

Refs:

- Breton S, Stewart DT, Shepardson S, Trdan RJ, Bogan AE, Chapman EG, Ruminas AJ, Piontkivska H, Hoeh WR. 2011. Novel protein genes in animal mtDNA: a new sex determination system in freshwater mussels (Bivalvia: Unionida)? Mol. Biol. Evol. 28:1645–1659.

- Lubośny M, Przyłucka A, Śmietanka B, Burzyński A. 2020. Semimytilus algosus: first known

hermaphroditic mussel with doubly uniparental inheritance of mitochondrial DNA. *Sci. Rep.* 10:11256.

Answer: We added the sentence: Recently, Lubośny and collaborators reported a first case of DUI in the marine hermaphroditic mussel *Semimytilus algosus*²¹

> Lines 63-65

"During early embryogenesis of DUI species, the ♂-type mitochondria are recognized and then eliminated in females. In males, however, ♂-type mitochondria persist and are aggregated in one blastomere destined to produce the male germline (i.e. gonad)."

This is the scheme of the DUI mechanism that has been used since the early days (1990s). New observations indicate that such mechanism should be revised. In a recent publication, in *Ruditapes philippinarum* we observed heteroplasmy at the organelle level in undifferentiated germ cells of both sexes and in male soma, whereas gametes were homoplasmic: eggs for the F-type and sperm for the M-type (Ghiselli et al. 2019, see below).

Ref:

- Ghiselli F, Maurizii MG, Reunov A, Ariño-Bassols H, Cifaldi C, Pecci A, Alexandrova Y, Bettini S, Passamonti M, Franceschini V, et al. 2019. Natural Heteroplasmy and Mitochondrial Inheritance in Bivalve Molluscs. *Integr. Comp. Biol.* 59:1016–1032.

> Lines 65-71

"This leads to the formation of heteroplasmic males with ♀-type mitochondria in all somatic tissues and ♂-type mitochondria in their gonads and sperm. DUI females are usually homoplasmic with only ♀-type mitochondria in somatic and gonadic cells. Even in DUI species, somatic cells of male and females are strictly homoplasmic with only ♀-type mitochondria present. Malfunctions in segregation and/or incomplete elimination of ♂-type mitochondria during early embryogenesis cause heteroplasmy in somatic tissues of single individuals, both male and female, in a small subset of DUI species."

This section contains too much generalization in my opinion. The presence of M-type in male somatic tissues is known to occur in *R. philippinarum* (Ghiselli et al. 2011, reference #22), *Venustaconcha ellipsiformis* and *Utterbackia peninsularis* (Breton et al. 2017, reference #14), and also in *Mytilus galloprovincialis* (see for example Obata et al. 2006 below). The same works reported also that M-F heteroplasmy is pretty common. The Authors are reporting a simplified scheme of the DUI system, as the most of us working on DUI have done for years. That said, evidence against that scheme is accumulating fast and in my opinion is now the time to acknowledge some deviations from the simplified mechanism reported here. In particular, the statements "DUI females are usually homoplasmic with only ♀-type mitochondria in somatic and gonadic cells" and "Even in DUI species, somatic cells of male and females are strictly homoplasmic with only ♀-type mitochondria present" we now know that are not true.

Answer: Thank you for the extensive comment, we re-wrote the section accordingly

Ghiselli et al. (2019) used immunohistostaining of maternal (F-type) and paternal (M-type) mitochondrial proteins across different stages of germ and somatic cell development in the Japanese carpet shell clam, *Ruditapes philippinarum*, to unravel the molecular momentum of sex differentiation in a DUI species. It resulted that eventual sex differentiation happens through a meiotic drive that eliminates M-type mitochondria in female germ cells ("late oocytes") and F-type

mitochondria in sperm cells. Staining showed primordial stem and germ cells to be heteroplasmic in both sexes, whereas post-meiotic germ cells are strictly homoplasmic for F-type (♀-type) mitochondria in females and M-type (♂-type) mitochondria in male clams. This leads to development of females which are usually homoplasmic with only ♀-type mitochondria in somatic and gonadic cells, whereas male *R. philippinarum*, commonly heteroplasmic in soma, sometimes with the ♂-type dominating the ♀-type mitochondria in different tissues, and exclusive ♂-type mitochondria in gonads and sperm. Only rarely females, heteroplasmic in somatic tissues, were encountered (Ghiselli et al. 2011). Presence of M-type in male somatic tissues is further known to occur in *Venustaconcha ellipsiformis* and *Utterbackia peninsularis* (Breton et al. 2017, reference #14), and also in *Mytilus galloprovincialis* (see for example Obata et al. 2006 below) and might be more common across DUI species than previously conceived.

In my opinion, the last sentence ("Malfunctions in segregation and/or incomplete elimination of ♂-type mitochondria during early embryogenesis cause heteroplasmy in somatic tissues of single individuals, both male and female, in a small subset of DUI species") is not sufficient to compensate for the generalization discussed above, because according to it, the observations deviating from the simple version of the mechanism are due to malfunction and are limited to "a small subset of species". I disagree with this view: we do not know which is the "rule" and which is the "exception", and since most of the DUI species have not been investigated thoroughly for the M/F-type distribution in different tissues, I would not say that the observed deviations pertain to only a small subset of species. We need to investigate a broader range of taxa to get a glimpse of the conserved features of the DUI mechanism, that is why this work is so important.

Ref:

Obata M, Kamiya C, Kawamura K, Komaru A. 2006. Sperm mitochondrial DNA transmission to both male and female offspring in the blue mussel *Mytilus galloprovincialis*. *Dev. Growth Differ.* 48:253–261.

Answer: Indeed when we re-wrote the paragraph, the Malfunctions sentence was eliminated .

> Lines 119-120

"The sites that distinguish the two mitochondria"
Divergent sites?

Answer: Corrected

> Lines 121, 123 and on

Gene/protein abbreviations: please use lower case italics for genes and upper case for proteins.

Answer: Corrected

> Line 123

Delete "at the DNA level".

Answer: Changes done

> Lines 126-128

"Although DUI species are frequently characterized by a 3' extension or duplication of their COX2 gene, a feature that is believed to be involved in the transmission of ♂-type mitochondria"

 "Although DUI species are frequently characterized by a 3' extension or duplication of their COX2

gene, a feature that some Author suggested to be involved in the transmission of ♂-type mitochondria"

Answer: Changes done

> Figure 2

Not a mandatory request at all, but as a fan of big maps of mt genomes, I would really like a larger (and clearer) map of M-and F-type mtDNA in *A. islandica* (such those obtained with Circos, OGDRAW, etc...).

Answer: We did the maps (Fig2)

> Lines 140-141

"we found that 300 male and female individuals from all sampled populations had the typical metazoan configuration with only ♀-type mitochondria in somatic cells (adductor muscle)."

Just a note: it is possible that some of such "typical" individuals carry M-type mtDNA in other somatic tissues (mantle, gill, digestive gland, foot...). I would not consider the adductor muscle as a proxy for all the somatic cells (see below)

Answer: We removed "somatic" and left "adductor muscle"

> Lines 141-142

"In 15 individuals, however, we detected only ♂-type mitochondrial DNA in somatic tissues." See above.

Answer: We removed "somatic" and substituted with "adductor muscle"

> Lines 175-179

"In other DUI species the ♂-type mitochondria, typically confined to the male gonads, are absent in somatic cells and, hence, do not occur in females at all. Cases of ♂-type mitochondrial "leakage", characterized by the co-presence of ♂- and ♀-type mitochondria in somatic tissue, have been reported for some DUI species, mostly in male but also female individuals"

These first statement is not true (please see comments for Lines 65-71) and the second is inaccurate (please see comments for Lines 23-25).

Answer: We read again the Zouros 2013 and indeed we made mistakes in our summary here... We rewrote like this:

Cases of mitochondrial heteroplasmy with ♂- and ♀-type mitochondria in somatic tissue have previously been reported for several DUI species. In the most extensively studied Mytilide family, some studies report consistent prevalence of M-type mtDNA (♂) in somatic tissues of males. Also in females over 50% of adductor muscle samples analyzed in the study of Obata et al (2011) were heteroplasmic, with random occurrence of M-type mtDNA in additional somatic tissues (e.g., foot, mantle). Moreover, in female somatic tissues the ♀-type was always predominant^{14,15,22-25}.

> Lines 193-195

"These results, in combination with the complete absence of ♀-type mitochondria with predominant ♂-type mitochondria in foot and adductor muscle (Fig. 4), rule out paternal leakage as an explanation of the observed pattern."

Two notes: 1) I am uncomfortable with statements of "complete absence" in a scientific paper; 2) I did not get the last part ("rule out paternal leakage as an explanation of the observed pattern").

Answer: We changed to : These results, in combination with the absence of detection of ♀-type mitochondria with predominant ♂-type mitochondria in foot, gills and mantle (Fig. 4), suggests the homoplasmic occurrence of the M-type genome in *A. islandica* somatic tissues.

> Line 218

What does "ng" stands for? I would expect a copy number or a ratio of some sort (e.g. with respect to housekeeping genes).

Answer: "ng" stands for "nanograms" [added for clarity]. Indeed here we measured the replicate number of cytochrome b; M-form (in ng) in relation to a standard of 28S + 18S rRNA (geometric means).

> Line 223

Given the experimental design I would refer to "transcription" rather than "expression". It's a long way from RNA to final product, especially in mitochondria.

Answer: corrected

> Line 224

"The number of active mitochondria per cell is linked to the cellular metabolic requirements of the organism and can be roughly estimated by the mtDNA copy number"

The mtDNA may roughly estimate the number of mitochondria (even if I am less and less convinced about this), but in my opinion cannot estimate the number of ACTIVE mitochondria. An inactive mitochondrion still has mtDNA (which is included in the copy number), but it does not have physiological activity... So I think the term "active" should be avoided.

Answer: corrected

> Lines 235-265

Recently, Sophie Breton and Collaborators published two very interesting papers about mitochondrial activity in DUI bivalves. Since these kind of studies are still quite rare in bivalves (especially those with a DUI focus), I think it would be very interesting if the Authors would comment their results in the light of those of Bettinazzi et al (2019, 2020, see below).

Refs:

- Bettinazzi S, Nadarajah S, Dalpé A, Milani L, Blier PU, Breton S. 2020. Linking paternally inherited mtDNA variants and sperm performance. *Philos. Trans. R. Soc. Lond. B Biol. Sci.* 375:20190177.

- Bettinazzi S, Rodríguez E, Milani L, Blier PU, Breton S. 2019. Metabolic remodelling associated with mtDNA: insights into the adaptive value of doubly uniparental inheritance of mitochondria. *Proc. Biol. Sci.* 286:20182708.

Answer: Thank you for the suggestion to comment on the papers by Bettinazzi and colleagues. We have included the references and have merged two paragraphs to discuss the issues in one place now.

Recent papers analyzed bioenergetics in spermatozoa, eggs, and gills of DUI species, including *A. islandica*, from the Northeast Atlantic, compared to “normal” bivalves relying strictly on maternal inheritance (SMI) (Bettinazzi et al 2019, 2020). A specific focus of these paper was on understanding the evolutionary benefit of M-type mitochondria in DUI sperm for male reproductive success (e.g. sperm motility and conservation of sperm that persists in the *offspring to start a “new mitochondrial generation”*). The authors documented “energy saving” functions of DUI sperm mitochondria which only use a smaller fraction of their maximum respiratory capacity (coupled /uncoupled respiration) for sustained motility compared to SMI sperm. Less excess COX activity over the upstream ETS system in DUI sperm mitochondria confers tight control of electron flow and speaks for a less reduced state of ETS intermediates and hence low ROS production compared to F-type mitochondria. In the interpretation of the authors, this supports better conservation of mitochondrial and mt-DNA integrity in sperm of DUI species. In a similar manner, our analysis indicates reduced OXPHOS capacity of ♂-type mitochondria in *A. islandica* somatic tissues (30% reduction of ETS and COX activity protein⁻¹) compared to the ♀-type mitochondria of “normal” specimens (note that we did not analyze mitochondrial isolates and cannot – at this stage - comment on mitochondrial respiration and coupling rates).

> Lines 266-275

I think this interpretation of the results is very interesting and indicates potential future investigations. That said, in my opinion the concepts should be expressed with a more dubitative style. The scenario hypothesized by the Authors is certainly possible, but I think it should be presented as a working hypothesis, not an established fact.

Answer: We agree and have changed the wording of the paragraph.

> Line 278

"The fact that these genuine underperformers"

I would not say that yet. The biochemical activity of M-type mitochondria might just be different, not worse.

Answer: We agree and have changed the sentence to: “The fact that animals carrying the ♂-mitotype in somatic tissues constitute... “

> Lines 277-305

This part is very interesting as well, and presents some intriguing questions. I am not yet convinced about the existence of a causal correlation between mt type and growth, because I can see the correlation, but of course a causal link needs further study. One of my concerns is that since mt type and sex are linked in most cases, I wonder if the observed correlation is not between mt type and growth, but between sex and growth. Many studies in bivalves have linked sex and growth, and in *R. philippinarum* we observed that females are usually bigger (heavier) than males.

That said, I appreciate the cautious tone the Authors used here, so I do not see any problem with some healthy speculation.

Answer: Thank you for the comment, we have intended this section as one possible interpretation.

> Lines 322-323

"Here we describe the first clear example in metazoans with a complete replacement of ♀-type by ♂-

type mitochondria in somatic tissue."

Again, this sounds a tad weird to me because M-type mtDNA (namely a mitochondrial lineage inherited through sperm and well separated by the egg-transmitted mitochondrial lineage) does not exist in species with strictly maternal inheritance. So in my opinion this sentence is technically true, but confusing.

Answer: We agreed: we changed to “ .. replacement of maternally inherited by paternally inherited mitochondria in somatic tissue.”

> Line 325

"The exclusive presence of ♂-type mitochondria in somatic cells, similar to the paternally inherited mtDNA in sperm might be caused by failure of the DUI specific selection, feminization of ♂-type mitochondria, or by the strict paternal inheritance of ♂-type mitochondria."

I find this sentence confusing. In particular:

"similar to the paternally inherited mtDNA in sperm"

The M-type in somatic tissues comes from the mtDNA carried by sperm. In what sense "similar"?

"feminization of ♂-type mitochondria"

Possible, but as far as I know there is no evidence yet of such a thing. As things stand now, it is not a parsimonious hypothesis.

"or by the strict paternal inheritance of ♂-type mitochondria"

I do not understand. M-type is strictly paternally inherited in DUI species...

Answer: We changed with : The predominant presence of ♂-type mitochondria in somatic cells might be caused by a failure of the DUI specific compartmentalization of mitochondria in the embryo

> Lines 325-327

"This unprecedented inheritance system places mitochondria in novel nuclear backgrounds and might afford adaptive advantages in novel or changing environments."

I disagree about this being an unprecedented inheritance system. The inheritance system is the DUI system, it is the amount of M-type in somatic tissues that is interestingly different and, in the case of individuals carrying mostly/only M-type mtDNA, unprecedented.

Answer: We have revised the whole sentence in the following manner in agreement with the reviewer:

This unprecedented case of individuals carrying only or mostly ♂-type mitochondria in somatic tissues represents a previously undescribed condition of mitochondrial-nuclear crosstalk, which may either be tolerable or even confer adaptive advantages in specific environments.

> Lines 348, 351, 353-355

The Authors used frozen tissues sampled in March 2010. This might originate some concerns. If properly stored, frozen tissues are still good for DNA extraction (even if, in my experience, the older they get, the more difficult is to get good DNA), so the DNA analyses should be fine. I have more concerns about RNA. In my experience it is really difficult to get good quality RNA from old samples. In this case, the approach is a RT-qPCR, so it is possible that the amplification of small fragments of RNA is unaffected by degradation, but I would take the results with a grain of salt. I am not an expert about mitochondrial activity assays so I don't know the impact that old samples could have had on the results.

All that considered, I do not think that this issue undermines the work which I think it is solid, but I would recommend to be cautious when presenting the results and drawing conclusions.

Answer: The RT qPCR was done in spring 2014 and the RNA quality was assessed prior to RT qPCR by running pure RNA on agarose gel to check the absence of degradation and the presence of 28S and 18S bands (which was the case, unfortunately, we did not keep the gel picture). Regarding the enzymes activities, they were realized in autumn 2014. Other measurements of these enzymes on older samples showed results in the same range (Rodriguez et al 2019, <https://doi.org/10.3389/fphys.2019.00946>).

> Line 456

"A between-groups principal analysis"

 "A between-groups principal component analysis"

Answer: Corrected

Reviewer #2

We thank this reviewer for his or her suggestions and would like to apologize for not responding to these comments in earlier submissions. We have now written a very thorough rebuttal to the points raised in the review which hopefully helps to clarify the paper.

"1. The "DUI aspect" of the paper

The authors claim that the two divergent mitotypes they sequenced in *A. islandica* represent the paternally- and maternally-derived mtDNAs typical of DUI species. The authors obtained their sequences from somatic tissues of different individuals from different populations. What surprises me is that they speculate on the fact that they represent DUI lineages when they provide no evidence at all that these two divergent mitotypes are associated with sex and gametes. Based on the results presented here, there is no indication that the two genomes are respectively paternally- and maternally-derived, actually neither enough evidence to claim that this is a species with DUI.

The main feature of a DUI system is that two mitochondrial lineages are strictly associated with anisogamous gametes, they coexist in the same populations and are species-specific. To claim that *A. islandica* is a DUI species, the sperm in ALL populations should bear the haplotype that the authors claim to be a paternally-derived mtDNA. The fact that this putative paternal haplotype is absent in almost all populations tested suggests that this is probably not the case. In their results and discussion section the authors wrote that they have previously successfully amplified the so-called σ^1 -type mtDNA in sperm of the same species (Dégletagne et al., 2016). However, this actually never happened. They did not check sperm, but male gonads, which, if we consider *A. islandica* as a DUI species, contain both haplotypes (gonads also contain somatic cells). There is no evidence at all that this "putative paternal mitotype" is transmitted by father to sons and, as such, there is no way to pretend it to be a paternally-derived lineage. Moreover, as males are undoubtedly present in all populations tested, so should be the here-called paternally-transmitted mtDNA."

Answer: This is an interesting point, however, it is beyond the scope of this paper, not the least for practical reasons as we and other research groups have not been able to reproduce this species in captivity.

In a previous paper we show that *Arctica islandica* is a species featuring doubly uniparental inheritance (DUI), based on mitochondrial cytb and 16s sequence differences, which was related to sex and tissue. We found a maternal-haplotype in different somatic tissue of all individuals and in reproductive tissue of female individuals only. We also found a paternal mitochondrial type in male reproductive tissue only, this was verified in 2 different *Arctica islandica* populations (Baltic Sea and North Sea, see Degletagne et al 2015, MBE). In summary, we have rewritten the manuscript and tried to be more specific in our phrasing showing which of the available evidence supports DUI in this species.

The existence/persistence of two sex-linked mitochondrial lineages is species-specific and cannot be population-specific for a DUI species. No matter the location of various populations, the genetic pool of e.g. *M. edulis* (and other DUI species) always contains both sex-linked mtDNA variants. In this paper, the so-called paternal haplotype was only found in 15 out of 315 individuals and restricted to somatic tissues of one (or two) specific population. How the authors would explain the existence of DUI in just one population (and even there, just 11% of the individuals bear the divergent mtDNA)? A DUI system is species-specific and never population-specific.

Answer: The empirical results are straightforward: Using the same methodology, our data show the presence of ♂-mitochondrial type in adductor muscle only in some populations but not in others. At present, we can only speculate as to the mechanisms that make this observation happen. We have tried to make it clear that different patterns of natural selection in different habitats might be responsible by disfavouring some mitochondrial genotypes in some habitats only (l. 164 -172) :

Considering the possibility that the ♂-type mtDNA might evolve faster with less strict purifying selection than the ♀-type mtDNA in DUI species¹⁸, it is possible that some cases of somatic ♂-type mitochondria escaped detection in our study if a mutation occurred at the binding sites of our ♂-type mtDNA specific primers. However, the successful amplification of only the ♂-type mtDNA in two different loci (cytochrome b and 16S) in sperm²⁹ and in somatic tissue (this paper) indicates that we can readily detect the ♂-type mtDNA if present. The population-specific occurrence of ♂-type mitochondria in somatic tissues could be the result of a strong selection against ♂-type mitochondrial DNA in individuals thriving in areas in which environmental conditions are more variable or alternatively reflect a founder effect even in the absence of selection.

Also, somatic tissues in DUI species, when they contain the paternally-derived mtDNA, are always heteroplasmic, and this is not the case here. This means that either the “divergent haplotype” found is maternally-inherited or that the methods used to detect both “putative DUI mitotypes” do not work. According to figure S5, the “putative paternal haplotype” has to be in low concentration to be able to detect the “maternal haplotype” using the multiplex PCR, so if it is in relatively high concentration, a heteroplasmic state cannot be detected. Also, the figure 3 is not really convincing, there is no detail about the “sex-specific regions” chosen – why not testing the entire mtDNAs? (actually this is what the authors wrote in the results section but according to the figure, this is not true).

Answer: In figure S5, we show that with our method we were able to detect the mitochondrial ♂-type at very low concentration, even with a high concentration of competing mitochondrial ♀-type present. Moreover, we sequenced the mitochondrial DNA entirely and found that the individuals show either one or the other haplotype. In figure 3 we only showed partial information but, as

mentioned in the text, we tested it on the overall DNA molecule because showing the full information would be difficult to overlook for the reader.

Moreover, we completed this analysis using qPCR assays, which detects in two different reactions each mitochondrial variant at very low concentration of the same DNA extraction or cDNA tube (down to 10^{-9} ng/ μ L of DNA) and we found no heteroplasmy for any individual carrying the ♂ mitochondrial type in somatic tissue.

The authors proposed that the absence of the paternally-derived in some populations could be the result of a strong selection against this “variant” in individuals living in areas characterized by a more variable environmental condition. In my opinion, it is more likely the opposite, as heteroplasmy could foster within population phenotypic plasticity (in this case, at least, as revealed by the OXPHOS effect), which could be better exploited in variable environments than in stable ones.

Answer: Thank you for this comment, it is indeed possible that mitochondrial heteroplasmy affords the individual with a higher degree of phenotypic plasticity, which is one of several competing explanations how natural selection might favour different mitochondrial configurations in populations across a vast distribution range. As this particular explanation would likely require to assume fluctuating environmental conditions to prevent one genotype being eliminated over evolutionary times, a possible and interesting scenario, we decided that at this stage it would take our speculation too far and presented but one, slightly more general scenario that would explain our observations.

In addition, the populations the authors studied are not so distant and share a pretty common environment. Even if the author proposed the environment to be very different, and I can agree that some differences surely exist, they did not compare populations living in different hemispheres, or Caribbean vs Arctic seas, but populations from Norway, Iceland, Baltic Sea, Helgoland, Kattegat, White sea etc. (all these individuals live in cold seas of the North hemisphere and they all live burrowed in sand at the bottom of the sea). For example, *R. philippinarum* (a DUI species) possesses the two “typic DUI lineages” no matter which population you test, and we are talking about populations of the Philippines, Italy, west coast of North America etc. The two mitotypes found in the present study might just represent two regional mitochondrial variants present within the same species, one widespread among the *A. islandica* populations and one underrepresented, specific of few individuals in the populations around Iceland.

Answer: We tested all populations across the distribution range in which *Arctica islandica* is present. The argument about the fact that the 2 mitochondrial DNAs could be regional variants is perfectly valid, but it is conspicuous that this variant can be identifiable as male mtDNA. This is now more explicitly described in lines # 111-115 of the corrected paper).

We also want to point out that Kiel Bight is an inland sea transitional zone and animals living there undergo massive fluctuation in term of salinity, amongst other, which results in huge consequences in term of physiology and longevity for *Arctica islandica*.

Overall, if *A. islandica* is a species with DUI, and if the two detected mitotypes are really the maternally- and paternally-derived mt lineages, they should be present in all the populations (and be strictly associated with sex and gametes), and this is not the case here (or has not been correctly tested).

Answer: See above, the observation has been carefully tested and published before, and the DUI effect is present in all male gonads of all populations within North Sea and Baltic Sea populations (Degletagne et al 2015). We concede that an evolutionary framework that would explain how this peculiar system evolved and through which steps is still missing.

The two mitotypes described could only reflect within species and between population mitochondrial heterogeneity found in somatic tissues of *A. islandica* individuals independently of sex (found in both females and males), and not associated with gametes. This cannot in any way be confused with the species-, sex- and gamete-specific association of all the DUI lineages described so far (in more than 100 species). As such, this paper is highly speculative (or incorrect).

Also, several complete maternal and paternal mt genomes of DUI species are present in GenBank and they all have something in common: they possess additional sex-specific protein-coding genes. Did the authors verify for the presence of such genes? The two mitotypes provided here are not so dissimilar (5.5% of divergence or less) compared to the standard DUI species divergence (from 20 up to 50%).

Answer: We did not verify the presence of sex specific protein coding genes, and we are not sure of what the reviewer refers to? the m- and f-ORF? As it was clearly not the point of the study, we did not investigate the ORF. The fact that the two mtDNA variants are not so dissimilar was also a surprise to us, but the data are what they are. (and see reviewer 1 suggestion on this point “ a recent split”)

2. The “mitochondrial functional aspect” of the paper

The authors claim that the expression of σ^1 -type mtDNA in somatic tissues reduces mitochondrial activity. Although interesting per se, the results presented here are in line with what has already been widely seen in several DUI species and do not provide anything new to the community other than supporting previous evidence. The phenotypic effect of bearing sex-linked mitochondria in DUI species has already been observed several times during the last decades. Starting from the effect of bearing male or female derived mitochondria upon sperm motility and bioenergetic strategy to the effect upon gametes and somatic tissue energy metabolism (Everett et al., 2004; Jha et al., 2008; Stewart et al., 2015; Bettinazzi et al., 2020). Concerning the OXPHOS, the activity of the two sex-linked mt lineages has been investigated (Milani and Ghiselli, 2015; Ghiselli et al., 2018) and even the depression in CCO activity was described ten years ago in *M. edulis* when comparing sperm carrying either the paternally-derived or the maternally-derived DUI haplotype (classic vs masculinized) (Breton et al., 2009). *Finally, in M. edulis and A. islandica (interestingly, the very same species of this research) an OXPHOS remodelling of M-type DUI mitochondria was recently described in both gametes and somatic tissues (Bettinazzi et al., 2019).*

Answer: We agree with this comment, we never said that our results are completely different from what was previously published with respect to energetics of the two mitochondrial types. The main difference is that our study described for the first time the ETS and CCO activities in *somatic tissue*

from individuals carrying only the ♂ mitochondrial type in their entire body (especially in *A. islandica*). That – we think – makes a difference, and for all we know it is a new finding.

Please also note that in the paper of Stefano Bettinazzi and colleagues it is only the *M. edulis* gills that tested positive for M-mtDNA but not the *Arctica* gills. If you take a look at Fig 3 of their paper, it is in *M. edulis* gills where specific differences are found between male/female gill mitos, but not *A. islandica* (see also written text on p. 5). The reason is probably very simply: those gills were not endowed with ♂ mitochondrial type. But in *Mytilus* and other species, heteroplasmic gills have been found. Therefore, new in our study is only the fact that some *A. islandica* are indeed homoplasmic (or mostly homoplasmic, see corrected version of our paper) for the M-type.

..... in *M. edulis* and *A. islandica* (interestingly, the very same species of this research)....

Answer: Please note that we did not study *Mytilus edulis*.

Concerning the experimental set up, it is unclear if the gills have been frozen before being weighted for the enzymatic analyses (according to the first paragraph of the methods no), so the correction for mg of tissue is probably not appropriate and activities should be corrected for mg of proteins instead.

Answer: For the biochemical measurement, as we did not use the entire tissue sample, we weighed the frozen tissue before homogenizing it, as usually done for biochemical measurements. All enzymes activities were measured from the same tissue homogenate. COX and ETS activities are still statistically different between individuals carrying the ♂ and ♀-type mitochondrial types when we corrected them with CS. However, as the CS activity is statistically different between male and female individuals, we thought that it would be confusing to show these ratio. Moreover, we did not measure protein content on these samples.

Also, the authors claim that **the metabolic changes presented here can influence female growth**. To sustain such hypothesis, you need to 1) provide a viable explanation on how this could be possible and 2) base yourself on more data. In my opinion, you need to provide a more exhaustive vision of cellular bioenergetics, not limited to ETS and CCO enzymes. A decreased activity of ETS (CI+CIII) and CCO is already known in DUI species and it would have been interesting having clues about the activity of other enzymes related with the energy metabolism. This includes enzymes forming part of the OXPHOS itself, as well as glycolysis, cytosolic fermentation pathways, fatty acid metabolism etc. It would have been interesting to test a panoply of enzymes, not only the two for which extensive evidence exists in the literature, in order to have a bigger vision of the effect upon cell energy metabolism. For example, concerning the OXPHOS it would have been nice to test CII. Since it is totally encoded by the nucleus, its activity would provide clues on whether the metabolic depression seen in CCO is really due by the interaction between a divergent mitotype and the common nuclear background.

Answer: Thank you, we tried to be more careful to designate what we consider a fact (mostly the correctness of our observations) and what is speculations about the underlying causes.

Regarding the CII complex activity, we tried to measure it but failed in our trials as more tissue would have been required for proper measurement. That is why we measured CS, another mitochondrial enzyme, not involved in OXPHOS but in Krebs cycle, encoded by the nucleus.

Dr Ghiselli (Rev. 1) asked us to further comment on the two papers by Bettinazzi *et al* which also deal with *A. islandica*, and to compare and associate these results with our present results. We have hence formulated an entire new paragraph which may answer some of your questions.

Finally, the authors are proposing that the replacement of one haplotype by the other in females might be evolutionary relevant and influence females *A. islandica* growth rates. Specifically, the authors claim: “Curiously, we found that females with somatic ♂-type mitochondria tend to have higher standardized residuals for all parameters tested, which might indicate that they are heavier and have a bigger and heavier shell than those with ♀-type somatic mitochondria.” I found this a clear overstatement. First, if evolutionary relevant, I would expect this condition to be more present at both within- and between-population levels. This putative replacement was not revealed in most population, and, when revealed, only 11% of the individuals was carrying it. Overall, saying that bearing a specific haplotype might be beneficial for females is highly speculative because:

- only 6 females carry the condition out of 315 total individuals;
- the authors base their claim on morphological parameters results which are not statistically significant (figure s4) and, even if they were, the n=6 is actually very small

Answer: As we had information regarding individual morphology, we tested this hypothesis, but we were really careful when analyzing the data by using “tend to” and “might” in our sentence. We agree that there is no possibility to conclusively and functionally test this, as we do not have enough samples (we were constrained by the number of individuals with the ♂ mitochondrial type we found amongst the 315 individuals we genotyped).

In the current version of the manuscript, we made an effort to more clearly distinguish between established findings and possible explanations, e.g.: “ However, at present such data have to be used with caution as the evidence is only based on six females with somatic ♂-type mitochondria, and these individuals do not fall outside the range of confidence intervals for LSG, AW or SW observed for the other individuals. »

As *A. islandica* is an animal that has raised interest because of its extreme lifespan to which no one can find a real clue, we think it relevant to make a statement about what our findings could mean for this longevity phenomenon. A key sentence in this whole argumentation is this one :

« We propose longevity extremes such as 507 years of lifespan⁴⁶ to require a very special disposition even in a centenarian species. »

Indeed it is an obvious thing that also in humans becoming a centenarian requires special conditions. Perhaps few animals featuring a 30% reduction of metabolism can, under specific conditions, lead a longer life in a « low demand environment » which allows for life on the slow lane. This is an idea that we propose. And we certainly never said that this could be « beneficial », but we said .. this could be favored in species reproducing to lives end ». We could remove this sentences if it would improve the paper (Line 370 – 372), but it highlights our ideas.

- How could the authors claim that the changes in activities of ETS and CCO has a female-specific effect on growth when their enzymatic measurements was not accounted for males and females separately (figure 5b,c)? i.e. the activities were calculated for all the sexes mixed together, not like citrate synthase (figure 5d). As such we cannot know whether the expressed mitochondrial phenotype might change among sexes (if there is a sex-specific phenotypic effect of bearing the specific haplotype).

Answer: We split the data to show the data and highlight the fact that there is no sex effect. In the first place, for ETS and CCO we compared only the mitotypes over all samples because there was no difference between sexes, but between mitotypes (and therefore within each sex). For CS we found a difference between sexes >> in short: females have less mitochondrial volume in gill tissue. When you then add both findings, females inheriting the less active paternal mitotype and present lower mitochondrial volume in gills they should definitely be the ones with less overall ETS and CCO (and currently unpublished results confirm this from mitochondrial isolates).

In response to Dr Ghiselli's comments we added a complete new paragraph from Line 296 – 328.

- How a lower activity of two ETS enzymes could explain a higher growth rate?
- Moreover, given that both males and females appear to bear this putative paternal haplotype in this population, why only females (and within females, only 6 of them) would be subjected to that?

Answer: For these 2 questions, we do not have any hypothesis and find it intriguing. We showed the data as they are. A vague idea is that the higher shell growth may be mediated through a potential change in burrowing behavior of these animals, but we really can't tell.

Overall, what is the metabolic property that allows only few female individuals to grow faster and why this haplotype is not widespread among populations? Also, the authors claim that “ σ^1 -type mitochondria, while not presenting an explicit advantage, are better tolerated at the high latitude by subarctic populations.” I am confused. For the authors, does this specific haplotype confer an advantage or not?

Answer: As we aimed to be careful with these intriguing data and cannot prove that this σ^1 mitochondrial type provides a clear advantage, we presented alternative hypotheses also potentially explaining the partition of the male mitochondrial variant.

The σ^1 mitochondrial type apparently is important for sperm motility and egg targeting in DUI species (see recent papers including Bettinazzi's). Here we look at an “accident” in the distribution of mitotypes across tissues presumable starting in the embryo. An accident must not necessarily represent an advantage, but under specific conditions, accidental mutations can survive. This is what we look at.

Moreover, the authors state: “Assuming this σ^1 -type mitotype is associated with a reduced ROS production, linked with the decrease of mitochondrial complex activities (Fig. 5B- 5C) in the mitochondria, this might even support extreme longevity in *A. islandica*.”

- How did the authors assume a reduced ROS production in this species if they did not measure neither ROS production nor the antioxidant activity?
- Where the individuals with complete replacement older than the others 300? Is there any indication that they would have lived longer? No.

On the contrary, in DUI species, the paternal-mitochondria activity appears to link with an increased ROS production (Monagan and Metcalfe, 2020; Milani and Ghiselli et al., 2015; Bettinazzi et al., 2019a). Also, as proposed by some studies on *A. islandica* (Munro et al. 2013; Blier et al., 2017;

Rodriguez et al., 2019), the “stoichiometric reorganization of OXPHOS components in *A. islandica* that might account for its low ROS production entail increasing the proportion of components upstream and downstream of the major ROS producing complexes (which are CI+III), that would allow reducing H₂O₂ efflux while maintaining total catalytic capacity.” By a fast look at figure 5 the stoichiometry between enzymatic complexes appears to be conserved, and not in line with a potential decreased ROS production associated with a putative paternal haplotype. Since there is no measure of ROS production (or evidence of changes in antioxidant activity), forcing a link between longevity in this species and the activity of just two enzymes is, in my opinion, completely speculative.

Answer: Agreed, it is always difficult to strike a balance between presenting facts only and becoming overly speculative in an attempt to highlight possible mechanisms at work. If this sentence is confusing, we can remove it. This point was just mentioned as a lot of studies are struggling with the fact that *Arctica islandica* can reach 504 years around Iceland, and that our study showed that this ♂ mitochondrial type is mostly found in the Icelandic population and other long-lived populations.

Reviewer #3 (Remarks to the Author):

Studies of DUI are especially interesting when they can provide new insights into strictly maternal inheritance itself. This study is useful for both the field of DUI and the study of mitochondrial inheritance by providing potential consequences of the maintenance (instead of removal) of male-type mitotypes. Although the study relies on a few exceptional individuals of the study species (which the authors have clearly stated), it will provide another direction for future studies of bivalves possessing DUI. Based on the methods provided I believe this type of study is repeatable with other bivalve populations.

With the above remarks in mind, I do have some specific questions and comments on the manuscript. Some larger questions I have:

(1) Were novel f- and m-orfs identified in individuals with complete mtDNA sequencing? To my understanding venerid species do not necessarily possess the f- and m-orf which are of interest to studies of mytilid and unionid DUI species.

Answer: We did not take a look to f- and m- orf as it was not the main point of our paper.

(2) Why was atp8 excluded from Figure 2? In other DUI possessing bivalves it can be difficult to identify, was this the case here?

Answer: Sorry, when I designed the figure, it was difficult to mention all genes clearly. I corrected this mistake by adding  and the  mtDNA maps in which ATP8 is indicated.

(3) In regards to line 416: Are these SNPs or SNVs? Are the variations represented in at least 1% of the population to qualify as a SNP? If so, is this supported by the paper?

Answer: We added these clarification in the material and methods section :

These differences were considered if, for one given position, nucleotides from all ♂ mitochondrial DNA sequences are different from nucleotides from all ♀ mitochondrial DNA sequences, considering gaps as a fifth nucleotide.

I believe a recent paper on two other venerid species would be of interest to this study (Capt et al. 2020 <https://doi.org/10.1038/s41598-020-57975-y>). It presents additional sequences that could supplement their tree (Figure 1) by introducing examples of DUI in two additional families (Solenioidea and Tellinoidea). It provides an interesting example of the cox2 extension the authors have discussed. Additionally, the findings may contribute to the discussion of divergence in lines 104-108.

Answer: We added these additional families in the tree.

Additionally, I have some specific comments:

Line 60: DUI is also found in Nuculanida (<https://doi.org/10.1007/s00227-012-2099-y> and [10.7717/peerj.2760](https://doi.org/10.7717/peerj.2760)).

Answer: Thank you, reference and taxon added

Line 60-61: It is not yet clear what percent divergence between mitotypes in DUI is “typical”, but there has been “lows” of 20% and “highs” of 50% documented. A reported range might be more clear to the reader, which the authors have done later (line 108).

Answer: Agreed, the sentence was rephrased accordingly. We did not mention *Arctica islandica* with 8% as it was only estimated on partial 16S and cytochrome b sequences.

In DUI species, the reported differentiation between the two mitochondrial lineages range from 20% in many marine taxa (orders Mytiloidea and Veneroidea) and can reach >50% in freshwater mussels (order Unionoidea) in their nucleotide sequences^{16,18,20,21}

Line 59/62: The authors use “dioecious” to describe separate sexes in a species, but later use “gonochoristic”, using one term consistently may benefit readers new to DUI research

Answer: Changed to “dioecious” for consistency throughout the MS

Although Fig. S2 and Fig. S3 are supplementary I believe they would benefit from the IDs being cleaned up for legibility or else locations be replaced by a colour legend, etc.

Answer: Done

Fig. S5. Would benefit from the ladder bands being labeled with their size, even though the products are labeled it would assist those who wish to repeat these methods.

Answer: Done

REVIEWERS' COMMENTS:

Reviewer #1 (Remarks to the Author):

I was one of the reviewers for the first submission of this work, and I gave a positive feedback, with some comments and suggestions.

The Authors performed an intensive revision of the manuscript, taking into account and addressing all my comments.

As I stated in my first review, I think works like this one are important and necessary to gather much needed information about the DUI system. Of course more thorough investigations are needed to clear some doubts (including some of Reviewer#2's concerns, which are legit), but I think these findings need to be published and shared with the community even if they are not conclusive (is ever a work "conclusive", anyway?). Just my two cents!

I am looking forward to more exciting work on *Arctica islandica* and other DUI bivalves.

Reviewer #2 (Remarks to the Author):

In the paper entitled "Presence of male mitochondria in somatic tissues and their functional importance at the whole animal level in the marine bivalve *Arctica islandica*" the authors report a special case of DUI in the ocean quahog. Specifically, the authors demonstrate that 1) some individuals of an Icelandic population appear homoplasmic for σ -type mitochondria in somatic tissues, 2) σ -type mitochondrial genes are transcribed, and 3) individuals with σ -type mitochondria in somatic cells lose a certain percentage of their wild-type respiratory capacity. The authors conclude that this situation could even confer adaptive advantages in specific environments. I think that the manuscript has been significantly improved, the authors responded to and sufficiently addressed the comments made by the reviewers. The paper should be accepted with minor revisions indicated below.

SPECIFIC COMMENTS

Abstract lines 18-19: "...whilst paternal mitochondria are typically eliminated in fertilized eggs" → while

Also, paternal mtDNA and mitochondria are sometimes eliminated in sperm before fertilization. Please consider rephrasing.

Abstract line 23: "...exclusive presence of σ -type mitochondria affects proteins partially encoded by mitochondrial genes and leads to a sharp drop in respiratory capacity".

This sentence is a bit confusing, why 'proteins partially encoded by mitochondrial genes'? I would delete this part of the sentence or clarify.

Abstract line 26: I thought males also lost their respiratory capacity?

Introduction line 60: "...tissues, and exclusive σ -type mitochondria in gonads and sperm".

M-type is exclusive only in sperm because gonads also contain somatic cells. Please correct.

Introduction line 75: "...sexes as well as in the gonads or females whereas the rarer.."

→ of females

Results line 159 please correct "heteroplasma" for heteroplasmy.

Results line 163 please add a dot at the end of the sentence.

Results line 216: "A specific focus of these paper was..."
-> these papers

Conclusion line 269 please correct in somatic tissues.

Conclusion line 271: "of individuals carrying only or mostly σ -type mitochondria in somatic tissues represents a previously undescribed condition of mitochondrial-nuclear crosstalk..."
Please add "in DUI species"

Materials and methods line 287: "...all tissues used in this study were frozen from previous studies..."
But in the results and also in line 368 it is claimed that enzymatic activities were "expressed in U.g-1 tissue fresh mass". Please clarify or correct.

Dear reviewers,

Thank you for your comments on our paper. It helps us to highly improve our manuscript. We corrected all points listed below.

Cyril Dégletagne, in the name of the entire authoring team.

REVIEWERS' COMMENTS:

Reviewer #1 (Remarks to the Author):

I was one of the reviewers for the first submission of this work, and I gave a positive feedback, with some comments and suggestions.

The Authors performed an intensive revision of the manuscript, taking into account and addressing all my comments.

As I stated in my first review, I think works like this one are important and necessary to gather much needed information about the DUI system. Of course more thorough investigations are needed to clear some doubts (including some of Reviewer#2's concerns, which are legit), but I think these findings need to be published and shared with the community even if they are not conclusive (is ever a work "conclusive", anyway?). Just my two cents!

I am looking forward to more exciting work on *Arctica islandica* and other DUI bivalves.

Best Regards,
Fabrizio Ghiselli, PhD

Thank you very much. Your previous comments help us a lot to improve our manuscript.

Reviewer #2 (Remarks to the Author):

In the paper entitled "Presence of male mitochondria in somatic tissues and their functional importance at the whole animal level in the marine bivalve *Arctica islandica*" the authors report a special case of DUI in the ocean quahog. Specifically, the authors demonstrate that 1) some individuals of an Icelandic population appear homoplasmic for ♂-type mitochondria in somatic tissues, 2) ♂-type mitochondrial genes are transcribed, and 3) individuals with ♂-type mitochondria in somatic cells lose a certain percentage of their wild-type respiratory capacity. The authors conclude that this situation could even confer adaptive advantages in specific environments. I think that the manuscript has been significantly improved, the authors responded to and sufficiently addressed the comments made by the reviewers. The paper should be accepted with minor revisions indicated below.

SPECIFIC COMMENTS

Abstract lines 18-19: "...whilst paternal mitochondria are typically eliminated in fertilized eggs"

→ while

Corrected

Also, paternal mtDNA and mitochondria are sometimes eliminated in sperm before fertilization. Please consider rephrasing.

Corrected : Metazoans normally possess a single lineage of mitochondria inherited from the mother (♀-type mitochondria) while paternal mitochondria are absent or eliminated in fertilized eggs.

Abstract line 23: "...exclusive presence of ♂-type mitochondria affects proteins partially encoded by mitochondrial genes and leads to a sharp drop in respiratory capacity". This sentence is a bit confusing, why 'proteins partially encoded by mitochondrial genes'? I would delete this part of the sentence or clarify.

Thank you for your comment, there was a confusion between mitochondrial protein and respiratory complexes. We rephrased the sentence : Exclusive presence of ♂-type mitochondria affects mitochondrial complexes partially encoded by mitochondrial genes and leads to a sharp drop in respiratory capacity.

Abstract line 26: I thought males also lost their respiratory capacity?

You are right, thank you for your comment. We removed "female" and consider all individuals

Introduction line 60: "...tissues, and exclusive ♂-type mitochondria in gonads and sperm". M-type is exclusive only in sperm because gonads also contain somatic cells. Please correct.

Corrected

Introduction line 75: "...sexes as well as in the gonads or females whereas the rarer.." → of females

Corrected

Results line 159 please correct "heteroplasma" for heteroplasmy.

Corrected

Results line 163 please add a dot at the end of the sentence.

Corrected

Results line 216: "A specific focus of these paper was..."

→ these papers

Corrected

Conclusion line 269 please correct in somatic tissues.

Corrected

Conclusion line 271: "of individuals carrying only or mostly ♂-type mitochondria in somatic tissues represents a previously undescribed condition of mitochondrial-nuclear crosstalk..." Please add "in DUI species"

Added

Materials and methods line 287: "...all tissues used in this study were frozen from previous studies..."

But in the results and also in line 368 it is claimed that enzymatic activities were "expressed in U.g-1 tissue fresh mass". Please clarify or correct.

Sorry for the mistake, it was "per tissue mass". As you noticed, all tissues were frozen.